## RESEARCH ARTICLE

# Stress-specific NONO interactomes reveal a key role of Hsp70 chaperone activity in regulation of paraspeckle formation

Isaac Odonkor[1], Birendra Kumar Shrestha[1,*], Stephanie Rose Nielsen[1,*], Athanasios Kournoutis[1], Ida Emilie Bjørlo[1], Saikat Das Sajib[1], Annica Hedberg[1], Toril Anne Grønset[1], Kenneth Bowitz Larsen[1], Jack-Ansgar Bruun[1], Erik Knutsen[1], Seyed Mohammad Lellahi[1] and Maria Perander[1,2,‡]

## ABSTRACT

Paraspeckles are stress-induced nuclear RNA–protein condensates that assemble on the long non-coding RNA *NEAT1*. Their increased formation under certain cellular circumstances has gained growing interest due to their association with serious human diseases, such as neurodegenerative disorders and cancer. The biological functions of paraspeckles still appear obscure, but increasing evidence suggests that they contribute to regulation of gene expression by recruiting specific proteins and RNA molecules. Here, we have characterized and compared two stress-enriched interactomes of the essential paraspeckle protein NONO in both wild-type and paraspeckle-deficient *NEAT1* knockout cells. We identified Hsp70 as part of stress-enriched NONO complexes in wild-type but not in *NEAT1*-depleted cells. We show that proteotoxic stress-induced paraspeckle formation and *NEAT1* expression are strictly dependent on Hsp70 chaperone activity. Our data demonstrate that both NONO and Hsp70 transiently translocate to the nucleolus during heat shock and that paraspeckle formation during recovery follows Hsp70-dependent relocation of NONO from the nucleolus to the nucleoplasm. Taken together, we demonstrate an important role of Hsp70 in paraspeckle assembly and identify a possible link between the nuclear protein quality control system and paraspeckles.

**KEY WORDS: Paraspeckles, *NEAT1*, NONO, Cellular stress responses, Hsp70, Heat-shock response, Proteostasis, TurboID**

## INTRODUCTION

RNA-containing biomolecular condensates, hereafter referred to as RNA condensates, have gained increasing attention over the past few years due to their important roles in regulating gene expression and maintaining cellular homeostasis, particularly when cells are exposed to stress (Roden and Gladfelter, 2021; Putnam et al., 2023; Hirose et al., 2023; An et al., 2021; Banani et al., 2017; Lyon et al., 2021; Sharp et al., 2022; Quinodoz et al., 2021; Han et al., 2024). Their formation through liquid–liquid phase separation is a universal phenomenon in cell biology and serves to concentrate specific proteins and RNA molecules into so-called membrane-less organelles (Shin and Brangwynne, 2017; Banani et al., 2017; Hyman et al., 2014). Several specific RNA condensates have been well-characterized both in the cytoplasm and in the nucleus. These include stress granules and P-bodies in the cytosol, and nucleoli, Cajal bodies, nuclear speckles and, more recently, paraspeckles, in the nucleus (Lafontaine et al., 2021; Sabari et al., 2020; An et al., 2021; Hirose et al., 2023). Recent research has demonstrated that RNA condensates, particularly stress granules, are important for proteostasis, and that their dynamic assembly and disassembly under specific cellular circumstances has a crucial role in preventing the formation of toxic protein aggregates (Ganassi et al., 2016; Mateju et al., 2017; Wang et al., 2020; Alberti and Hyman, 2021; Amer-Sarsour and Ashkenazi, 2019; Han et al., 2024; Akaree et al., 2025). This is particularly important in neurons, and many serious neurodegenerative diseases are associated with formation of pathologic protein aggregates, which eventually will lead to cell death (Alberti and Hyman, 2021; de Oliveira et al., 2019; Sala et al., 2017; Han et al., 2024).

Cells are constantly exposed to stress. To counteract and adapt to this, several stress response pathways exist that are conserved in evolution. A major function of such pathways is to promote re-programming of gene expression, so that genes that are specifically required to preserve homeostasis are turned on, whereas many other genes are temporarily shut off (Himanen and Sistonen, 2019). One such stress response mechanism is the heat-shock response, which is elicited when cells are exposed to proteotoxic stress such as high temperature or agents that alter protein folding (Gomez-Pastor et al., 2018; Joutsen and Sistonen, 2019). The heat-shock response is orchestrated by the transcription factor heat shock factor 1 (HSF1), which turns on the expression of genes that are critical for maintaining proteostasis. The most well-described target genes of HSF1 are those encoding protein chaperones, such as members of the Hsp70 family. Hsp70 proteins (see below, hereafter Hsp70) play crucial roles in virtually all aspects of proteostasis spanning from housekeeping functions, such as ensuring proper folding, subcellular localization, activity and turnover of cellular proteins, to counteracting proteotoxic stress by promoting refolding or degradation of misfolded proteins via the ubiquitin-proteasome system (UPS), and by facilitating disentangling of proteins from aggregates (Mogk et al., 2018; Rosenzweig et al., 2019; Wentink and Rosenzweig, 2023; Mayer and Bukau, 2005). As such, Hsp70 plays a crucial role in preventing accumulation of potentially toxic protein aggregates (Mogk et al., 2018; Rosenzweig et al., 2019). More recently, it has been demonstrated that Hsp70 activity is instrumental for the dynamic assembly and disassembly and biophysical properties of RNA condensates (Ganassi et al., 2016; Lu et al., 2022; Yu et al., 2021). Hsp70 activity is regulated by ATP binding and hydrolysis that coordinate polypeptide substrate binding and release. This intricate cycle is dependent on and closely regulated by two major classes of

[1]Department of Medical Biology, Faculty of Health Sciences, UiT the Arctic University of Norway, 9037 Tromsø, Norway. [2]Centre for Clinical Research and Education, University Hospital of North Norway, 9038 Tromsø, Norway.
*These authors contributed equally to this work

‡Author for correspondence (maria.perander@uit.no)

co-chaperones – the J-domain proteins (JDPs), which facilitate substrate binding and ATP hydrolysis, and nucleotide exchange factors (NEFs), which promote ADP-ATP exchange required for substrate release (Kampinga et al., 2019; Bracher and Verghese, 2015). There are 13 Hsp70 members in humans, which are encoded by the *HSPA* genes. These include compartment-specific members, such as BiP (*HSPA5*) and mtHsp70, also known as mortalin (*HSPA9*) with housekeeping functions in the ER and mitochondria, respectively, and the more versatile members of which the constitutively expressed Hsc70 (*HSPA8*) and the stress-induced Hsp70 (encoded by the closely related *HSPA1A* and *HSPA1B* genes) are by far the most well studied (Kampinga et al., 2009).

Paraspeckles are nuclear RNA condensates formed around an architectural long non-coding RNA called nuclear paraspeckle assembly transcript 1 (*NEAT1*) (Clemson et al., 2009; Fox et al., 2002, 2018). *NEAT1* consists of two overlapping isoforms, *NEAT1_1* of 3.7 kb and *NEAT1_2* of 22.7 kB, which are transcribed from the same promoter (Sunwoo et al., 2009; Sasaki et al., 2009). Whereas *NEAT1_2* is essential for paraspeckle assembly, little is known about the functions of *NEAT1_1* even though this isoform is more broadly expressed across different cell types and tissues compared to *NEAT1_2* (Nakagawa et al., 2011; Adriaens et al., 2019; Li et al., 2017). *In vivo*, *NEAT1_2* seems to be most highly expressed in secretory cells within the stomach, corpus luteum and mammary gland (Adriaens et al., 2019; Nakagawa et al., 2014; Standaert et al., 2014). *NEAT1_2* is rarely expressed in stem cells, which consequently lack paraspeckles, but is induced during differentiation in certain developmental processes (Chen and Carmichael, 2009; Modic et al., 2019; Hupalowska et al., 2018; Sunwoo et al., 2009). Most established cell lines, however, express high levels of both *NEAT1_1* and *NEAT1_2* (Nakagawa et al., 2011). *NEAT1_2* expression and paraspeckle formation are elevated in response to a wide variety of cellular stress, and enhanced paraspeckle formation appears to be a global stress response mechanism in mammalian cells (Choudhry et al., 2015; Hirose et al., 2014; Adriaens et al., 2016; Lellahi et al., 2018; McCluggage and Fox, 2021; Wang et al., 2018). Emerging evidence suggests that paraspeckles increase cell survival during stress. The most prominent and well-established function of paraspeckles is to sequester proteins and RNA molecules, and through this, regulate gene expression at different levels (Hirose et al., 2014; Imamura et al., 2014; Chen and Carmichael, 2009). More recently, paraspeckles have been shown to coordinate transcription of specific groups of genes by physically linking their genomic regions (Toya et al., 2024). Many proteins are associated with paraspeckles, of which some are essential for their assembly and integrity, whereas others do not seem to play a role in their formation per se (Fox et al., 2005, 2002; Naganuma et al., 2012; Yamazaki and Hirose, 2015). The members of the *Drosophila* behaviour/human splicing (DBHS) family, NONO, SFPQ and PSPC1, are the most well-characterized paraspeckle-associated proteins (Fox et al., 2005, 2002, 2018; Dong et al., 1993). They are versatile proteins that form homo- or hetero-dimers that can regulate many aspects of RNA metabolism as well as DNA repair (Knott et al., 2016). Paraspeckle assembly is nucleated by binding of NONO or SFPQ polymers to newly synthesized *NEAT1_2* transcripts, and consequently, NONO and SFPQ are, together with FUS, RBM14, HNRNPK, HNRNPH3 and DAZAP1, essential for their formation (Fox et al., 2005, 2002; Naganuma et al., 2012; Yamazaki and Hirose, 2015). As is the case for other RNA condensates, paraspeckles assemble through liquid–liquid phase separation, which is driven by intrinsically disordered low-complexity domains, which are present in many paraspeckle-associated proteins including FUS, RBM14 and

the DBHS proteins (Yamazaki et al., 2018; Hennig et al., 2015; Franzmann and Alberti, 2019).

Here, we have explored and compared starvation- and sulforaphane-(SFN)-induced NONO interactomes in wild-type and paraspeckle-deficient *NEAT1*-knockout cells by quantitative proximity proteomics. We demonstrate that Hsp70 becomes enriched in stress-induced NONO complexes in wild-type but not in *NEAT1*-depleted cells. Proteotoxic stress-induced paraspeckle formation and *NEAT1_2* expression are strictly dependent on Hsp70 chaperone activity. Both Hsp70 and NONO transiently relocate to the nucleolus in response to heat stress and then return to the nucleoplasm during recovery, which precedes paraspeckle formation. The redistribution of Hsp70 and NONO to the nucleoplasm is severely hampered by inhibition of Hsp70 activity. We propose that Hsp70 is required for the generation of a nucleoplasmic pool of NONO that can be engaged in *de novo* paraspeckle assembly during recovery from proteotoxic stress.

## RESULTS
### Identification of stress-specific NONO interactomes
There is mounting evidence that upregulation of paraspeckle formation is a general cellular response mechanism to extrinsic and intrinsic stress. However, to our knowledge, no one has systematically compared the protein composition of paraspeckles formed in response to different stressors. We therefore decided to investigate by quantitative proximity proteomics, the interactome of the essential paraspeckle protein NONO in cells that were either treated with sulforaphane (SFN) (Lellahi et al., 2018) or subjected to starvation in Hank's balanced salt solution (HBSS), two conditions that induce *NEAT1* expression and paraspeckle formation in MCF7 cells (Fig. S1). To specifically identify paraspeckle-dependent partners of NONO, we included both wild-type cells and *NEAT1*-deficient cells in the study. First, we generated a MCF7 *NEAT1*-knockout (N1_KO) cell line by deleting the entire *NEAT1* locus by CRISPR/Cas9 genome editing (Fig. 1A). Mapping of RNA-seq data and Real-time quantitative RT-PCR (RT-qPCR) analyses confirmed that *NEAT1* expression was completely abrogated in the N1_KO cells (Fig. 1A, Fig. S2). As expected, as *NEAT1_2* is an essential scaffold for paraspeckle assembly, *NEAT1*-deficient cells did not form paraspeckles upon SFN-treatment or starvation, as opposed to what was seen in wild-type cells (Fig. 1B). We then generated stable cell lines of wild-type and N1_KO cells that ectopically expressed a fusion protein of NONO and the biotin ligase TurboID containing a V5 epitope tag (TurboID–V5–NONO), from a doxycycline (DOX)-inducible promoter (Fig. 1C). Western blot analyses using antibodies recognizing either the V5 tag or the NONO protein demonstrated that DOX induced the expression of a protein corresponding to the expected size of the fusion protein in both cell lines (Fig. 1C). The expression level of TurboID–V5–NONO was low compared to endogenous NONO, making it less likely to interfere with the stoichiometry between NONO and its cellular protein partners. Importantly, TurboID–V5–NONO colocalized with *NEAT1_2* and the paraspeckle marker protein PSPC1 in punctuated structures in wild-type MCF7 cells, confirming that it can localize to paraspeckles (Fig. 1D,E). In contrast, TurboID–V5–NONO displayed diffuse localization in the N1_KO cell line (Fig. 1F). Finally, we demonstrated that the TurboID moiety of the fusion protein is active, as addition of DOX promoted biotinylation of proteins in cells incubated in the presence of biotin (Fig. 1G).

We then conducted a proteomic experiment designed to identify (1) stress-specific NONO interactomes and (2) *NEAT1*-dependent stress-induced NONO interaction partners (Fig. 2A).

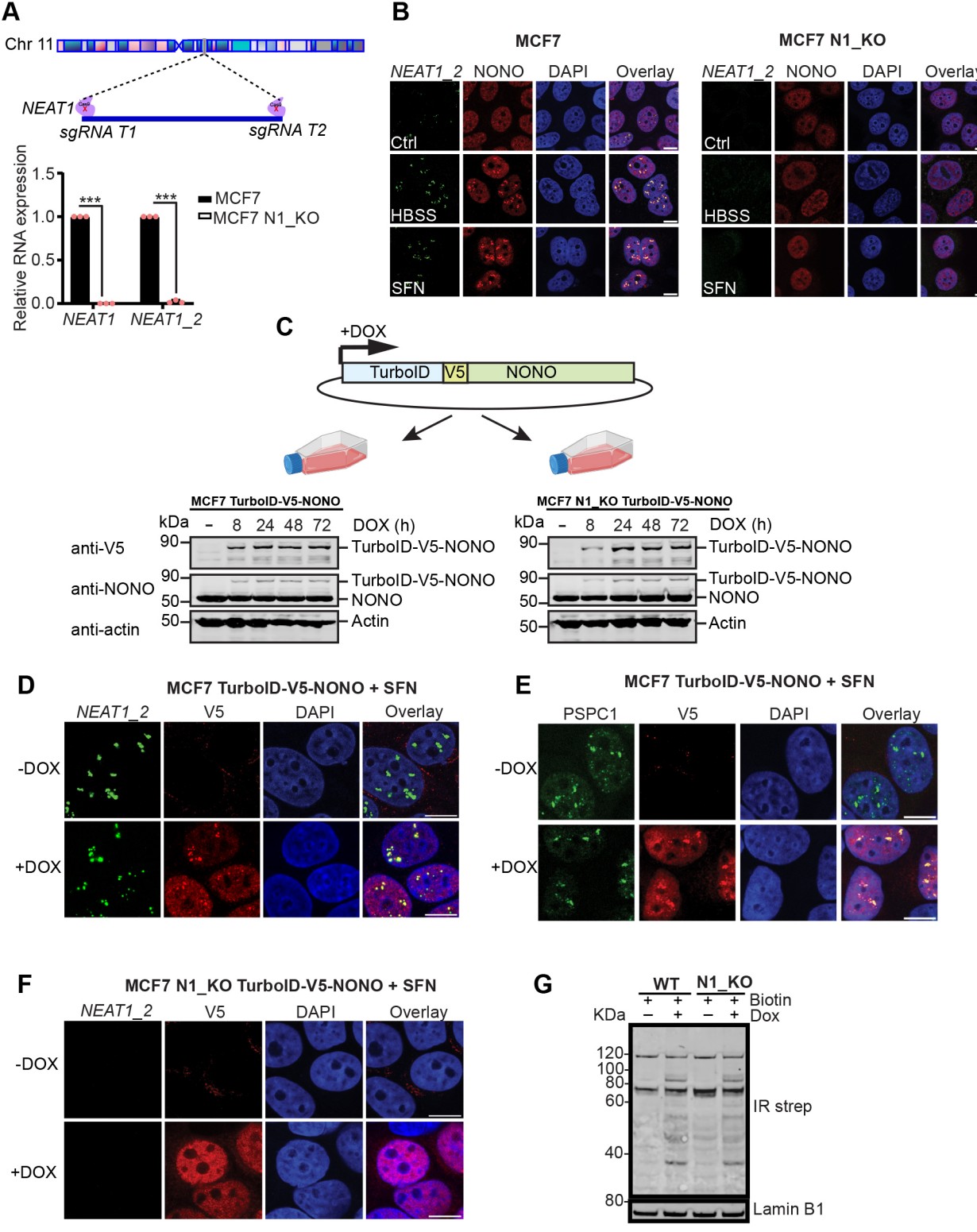

**Fig. 1.** See next page for legend.

TurboID–V5–NONO-expressing wild-type and N1_KO cells were incubated for 6 h in the presence of SFN or in medium depleted of serum and amino acids, and nuclear extracts were isolated from both stressed and control cells grown in normal conditions. Biotinylated proteins were affinity purified from the nuclear extracts using streptavidin-coated beads, and enriched proteins from four independent experiments were analyzed by tandem mass tagging (TMT) quantitative proteomics. In the following analyses of the proteomic data, we specifically focused on the NONO interactome that was enriched during SFN treatment or starvation compared to cells cultivated in normal conditions. For SFN-treated cells, we identified 58 and 305 significantly enriched NONO interactors in

**Fig. 1. Generation of wild-type and *NEAT1*-knockout cell lines with doxycycline-inducible expression of TurboID–V5–NONO.** (A) *NEAT1* expression is completely abrogated in a MCF7 *NEAT1*-knockout cell line (MCF7 N1_KO). The entire *NEAT1* locus was deleted by CRISPR/Cas9 genome editing. The relative expression of *NEAT1* (*NEAT1_1*+*NEAT1_2*) and *NEAT1_2* in MCF7 and MCF7 N1_KO cells was determined by RT-qPCR. *NEAT1* expression was normalized to GAPDH expression. The mean± s.d. of three replicates is presented. \*\*\*$P \leq 0.001$ (unpaired two-tailed Student's *t*-test). (B) MCF7 N1_KO cells do not form paraspeckles in response to starvation (HBSS) or sulforaphane (SFN). MCF7 and MCF7 N1_KO cells were treated with SFN or HBSS for 6 h and subjected to *NEAT1_2* (green channel)- and NONO (red channel)-specific co-immuno-FISH analyses by confocal microscopy. The nuclei are visualized by DAPI staining (blue channel). (C) Generation of MCF7 and MCF7 N1_KO cell lines that express TurboID–V5–NONO from a doxycycline (DOX)-inducible promoter. DOX-induced expression of TurboID–V5–NONO at different time points was determined by immunoblotting using antibodies recognizing the V5 tag or NONO. Equal loading was verified by re-probing the membranes with an anti-actin antibody. Part of this panel was created in BioRender by Perander, M., 2026. https://BioRender.com/e134t2b. This was sublicensed under CC-BY 4.0 terms. (D,E) TurboID–V5–NONO localizes to paraspeckles. MCF7 TurboID–V5–NONO cells were incubated in the presence of SFN for 6 h. Localization of TurboID–V5–NONO to paraspeckles was visualized by *NEAT1_2* (green channel)- and V5 (red channel)-specific co-immuno-FISH analyses (D) or co-immunofluorescent staining using antibodies towards PSPC1 (green channel) and V5 (red channel) (E). Nuclei are visualized by DAPI staining (blue channel). (F) TurboID–V5–NONO localizes diffusely in the nucleoplasm of MCF7 N1_KO TurboID-V5-NONO cells as assessed by *NEAT1_2*- and V5-specific co-immuno-FISH analyses as described above. (G) TurboID–V5–NONO has biotin ligase activity. MCF7 TurboID–V5–NONO and MCF7 N1_KO TurboID–V5–NONO cells were incubated in the absence or presence of DOX for 24 h to turn on TurboID–V5–NONO expression. Cells were treated with biotin for 30 min and nuclear extracts were harvested, separated by SDS-PAGE and blotted on to a nitrocellulose membrane. Biotinylated proteins were visualized by incubating the membrane in the presence of streptavidin conjugated to an infrared (IR) fluorescent dye. Equal loading was verified by re-probing the membranes with an anti-lamin B1 antibody. WT, wild type. Images in B–G are representative of at least three repeats. Scale bars: 10 µm.

the wild-type and N1_KO cells, respectively (Fig. 2B; Table S1). The low number of enriched proteins in the wild-type cells compared to N1_KO cells, is most likely a consequence of SFN-induced reduction in TurboID–V5–NONO levels in the wild-type cells, as lower amounts of the TurboID protein were detected by mass spectrometry (MS) in these samples (Table S1). Following starvation, 134 and 232 significantly enriched NONO partners were identified in the wild-type and N1_KO cells, respectively (Fig. 2C; Table S1). To start deciphering SFN-specific and starvation-specific NONO interactomes, we inspected the list of proteins that were enriched after exposure to these two cellular stressors in wild-type cells, N1_KO cells or in both cell lines. Among the stress-induced NONO interactors, we found 173 and 139 proteins to be unique for SFN-treated and starved cells, respectively, whereas 148 proteins were recruited to NONO during both conditions (Fig. 2D). Pathway enrichment analysis demonstrated that both SFN- and starvation-induced NONO-interactors were enriched in proteins associated with 'RNA metabolism', 'Ribonucleoprotein complex biogenesis' and 'Ribosomal large subunit biogenesis' (Fig. 2E). Generally, many of the proteins that are associated with these gene ontology (GO) terms are nucleolar proteins, indicating that NONO moves to the vicinity of, or into the nucleolus, during proteotoxic stress or starvation. We also identified pathways that were specific for the two stressors. For instance, proteins associated with 'Cell cycle' were only enriched in SFN-treated cells, whereas proteins associated with 'Integrator complex' were only enriched during starvation.

## NONO associates with Hsp70 in response to stress

Next, we sought to identify proteins that were recruited to NONO during stress in a manner that was dependent on paraspeckles. We compared the stress-induced NONO interactomes in wild-type and N1_KO cells, separately focusing on each of the two stressors. Among the 58 NONO-interacting proteins that were enriched during SFN treatment in wild-type cells, we identified 16 proteins that were seemingly dependent on *NEAT1* expression (Fig. 3A; Table S1). During starvation, we identified 55 proteins that associated with NONO uniquely in wild-type cells (Fig. 3B; Table S1). Finally, we compared the paraspeckle-dependent SFN- and starvation-induced NONO interactomes, and found two shared proteins, the HSPA1A/HSPA1B (HSPA1A and HSPA1B, which encode Hsp70 proteins that only differ by one amino acid)-encoded heat shock protein 70 (Hsp70) and Nucleolar protein 56 (NOP56) (Fig. 3C). The *HSPA1A* and *HSPA1B* genes encode two nearly identical stress-induced Hsp70 members that are not distinguished in the proteomic data.

Paraspeckle-dependent recruitment of the Hsp70 chaperone particularly caught our attention as (1) we have previously shown that *NEAT1* and paraspeckle formation are induced by SFN via the heat-shock response (Lellahi et al., 2018), (2) Hsp70 has been shown to play an important regulatory role in RNA condensate formation (Ganassi et al., 2016; Mateju et al., 2017), and (3) Hsp70 has a key role in proteostasis of nuclear proteins (Nollen et al., 2001; Frottin et al., 2019). As *HSPA1A/HSPA1B* are HSF1 target genes that are potently induced by SFN, we verified by RT-qPCR that the *HSPA1A* mRNA is equally increased by SFN in the wild-type and *NEAT1*-depleted cells (Fig. S3A). In line with this, the HSF1 transcription factor was activated by SFN to a similar extent in the two cell lines, as demonstrated by a phosphorylation shift in western blot analyses (Fig. S3B). Finally, we show that the endogenous Hsp70 protein levels were similar in wild-type and the *NEAT1* knockout cells (Fig. S3B). Of note, we could not detect increased expression of the Hsp70 protein after 6 h incubation with SFN in any of the cell lines. One explanation could be that a longer incubation period is needed to see any major changes in the protein levels in response to SFN in MCF7 cells.

To see whether NONO and Hsp70 could form complexes in another cell line, we transiently co-transfected HeLa cells with plasmids expressing GFP–Hsp70 (*HSPA1A*) and Myc epitope-tagged NONO. Myc-tagged, as well as endogenous NONO, co-immunoprecipitated with GFP–Hsp70 (Fig. 3D). Moreover, ectopically expressed Hsp70 formed complexes with other key paraspeckle-associated proteins including SFPQ, and its already well-defined partner, TDP-43 (also known as TARDBP) (Gu et al., 2021; Yu et al., 2021). In this experimental set up, we did not see a further increase in complex formation between Hsp70 and the paraspeckle-associated proteins upon SFN incubation.

## Paraspeckle formation is dependent on Hsp70 activity

Hsp70 and other chaperones have previously been shown to play instrumental roles in the regulation of biophysical properties of both cytoplasmic and nuclear RNP bodies, and through this, preventing RNA-binding proteins from forming pathological aggregates during cellular stress (Ganassi et al., 2016; Mateju et al., 2017; Agnihotri et al., 2025). Many paraspeckle-associated proteins, as well as many other nuclear proteins, possess intrinsically disordered low-complexity regions, which make them particularly prone to form aggregates. To start unraveling whether Hsp70 regulates paraspeckle integrity during proteotoxic stress, we incubated HeLa, MCF7 and U2OS cells with the Hsp70 inhibitor VER-155008 before inducing the heat-shock response by SFN, and visualized paraspeckles by

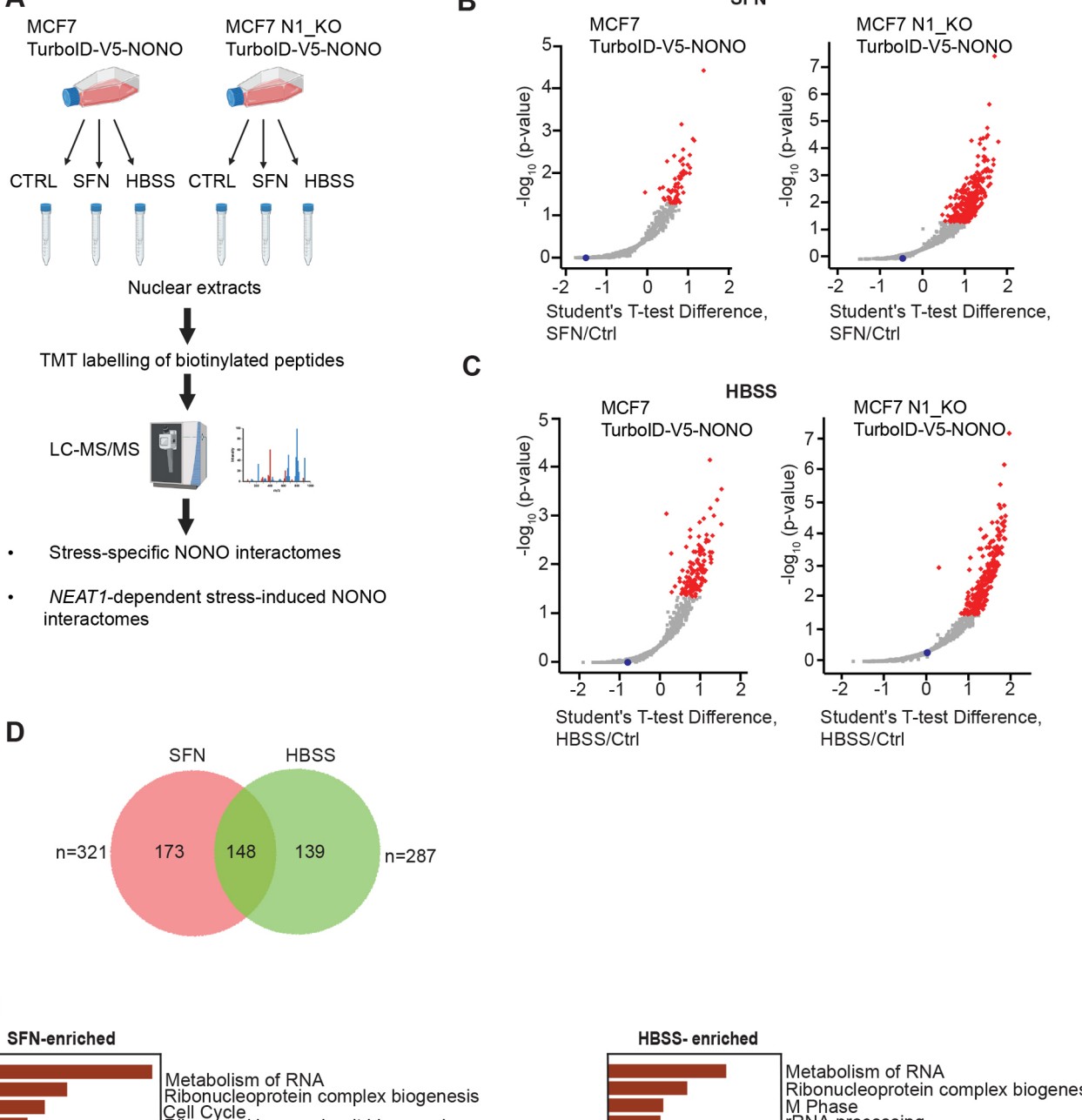

**Fig. 2. Identification of SFN- and starvation-enriched NONO-interactomes in wild-type and *NEAT1*-knockout cells.** (A) Schematic representation of the study outline. Nuclear extracts were harvested from control (CTRL), SFN- and HBSS-treated MCF7 TurboID-V5-NONO and MCF7 N1_KO TurboID-V5-NONO cells. Biotinylated proteins were purified using streptavidin-conjugated magnetic beads and subjected to tandem mass tag (TMT) labeling and liquid chromatography-tandem mass spectrometry (LC-MS/MS). Data analyses were performed to identify stress-specific NONO interactomes and *NEAT1*-dependent stress-induced NONO interactomes. Created in BioRender by Perander, M., 2026. https://BioRender.com/xlj9ep. This was sublicensed under CC-BY 4.0 terms. (B,C) Scatter plots showing SFN (B) and HBSS (C)-enriched NONO interacting partners in wild-type and *NEAT1*-knockout (N1_KO) cells. The blue dots correspond to the TurboID protein (denoted 'bioid' in Table S1). Mass spectrometry data were analyzed using the MaxQuant and Perseus software. Statistics were calculated from four independent experiments using the unpaired two-tailed Student's *t*-test. Significantly enriched proteins are highlighted in red (fold change 2, *P*<0.05). (D) Venn diagram showing shared and stress-specific partners of NONO in MCF7 wild-type and *NEAT1* knockout cells. (E) Pathway enrichment analyses of proteins that were significantly enriched in NONO-complexes in response to SFN and HBSS. Images are from pooled data from four independent experiments.

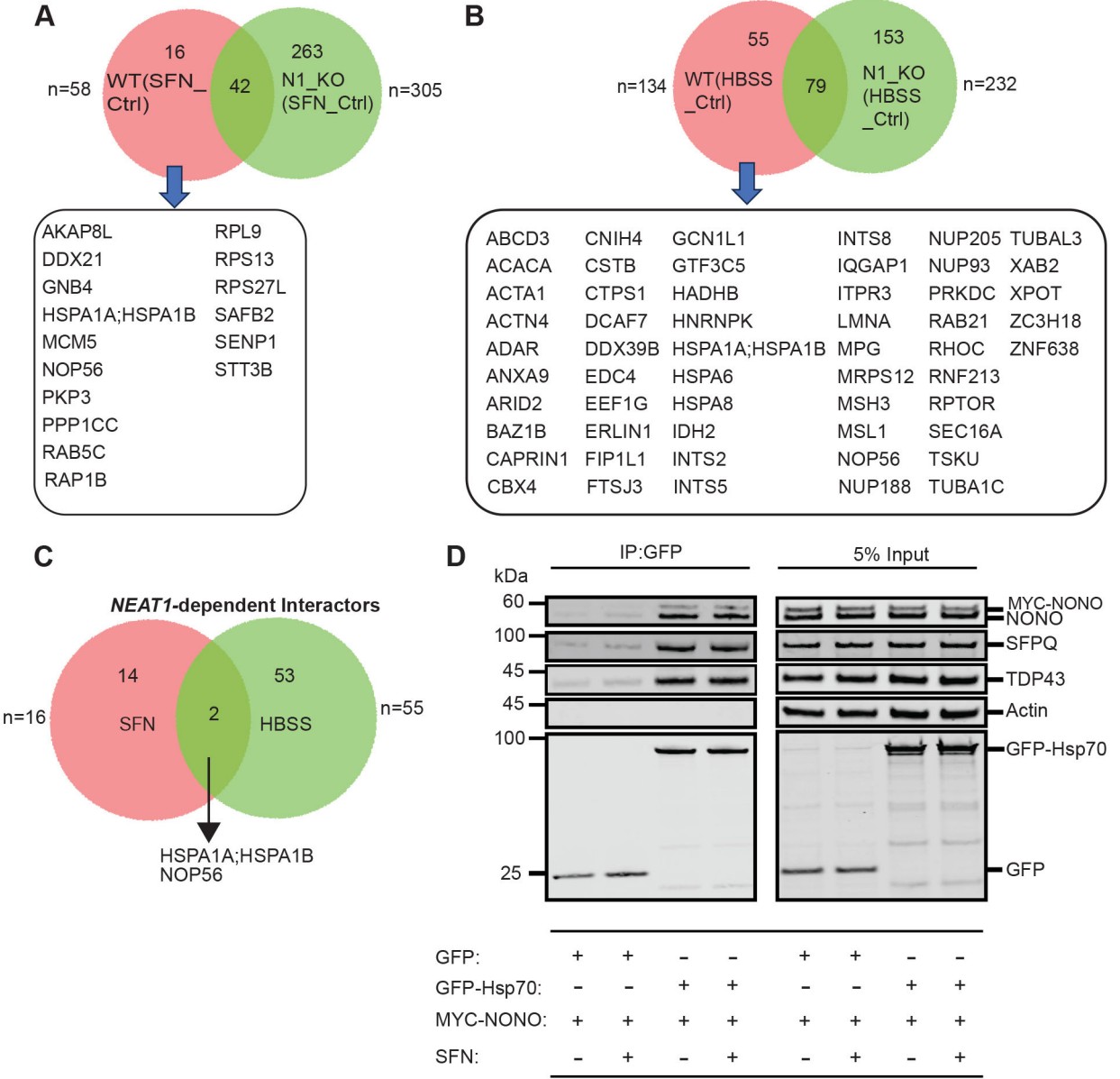

**Fig. 3. *NEAT1*-dependent enrichment of Hsp70 in SFN- and HBSS-induced NONO complexes.** (A) Venn diagram showing unique and shared proteins that become enriched in NONO complexes upon SFN treatment of wild-type (WT) and *NEAT1*-knockout (N1_KO) cells. *NEAT1*-dependent SFN-induced NONO partners are listed. (B) Venn diagram showing unique and shared proteins that become enriched in NONO complexes upon starvation (HBSS) of wild-type and *NEAT1*-knockout cells. *NEAT1*-dependent HBSS-induced NONO partners are listed. (C) Hsp70 and NOP56 are enriched in both SFN- and HBSS-induced NONO complexes in wild-type but not in *NEAT1*-knockout cells. (D) NONO, as well as other paraspeckle-associated proteins, interact with ectopically expressed Hsp70. HeLa cells were transiently co-transfected with plasmids expressing Myc-tagged NONO and either GFP or GFP–Hsp70. At 24 h post transfection, cells were either left untreated or treated with SFN for 6 h and protein extracts were harvested. GFP or GFP–Hsp70 was immunoprecipitated using GFP-Trap® Agarose (IP). Co-immunoprecipitated Myc-tagged and endogenous NONO, endogenous SFPQ and TDP-43 were detected by western blot analyses. Actin was not co-immunoprecipitated with Hsp70 and was regarded as a negative control. The expression of proteins in cell lysates corresponding to 5% of the input in the immunoprecipitation experiments is shown on the right. The results are representative of three independent experiments.

NONO- and *NEAT1_2*-specific co-immuno-fluorescence *in situ* hybridization (FISH) analyses. Intriguingly, inhibiting Hsp70 activity severely compromised SFN-promoted paraspeckle formation in HeLa and U2OS cells, and, when increasing the concentration by twofold, also in MCF7 cells (Fig. 4A,B). In HeLa cells, we also observed a clear reduction in the basal number of paraspeckles upon adding VER-155008 to cells cultivated in normal conditions (Fig. 4A,B). We repeated the experiment in HeLa cells using another Hsp70 inhibitor, HS-72, which had similar effect to VER-

155008 in reducing paraspeckle formation in both control and SFN-treated cells (Fig. S4A,B). We then went on to determine whether inhibiting Hsp70 activity also affected the expression of *NEAT1*. In line with its effect on paraspeckle formation, VER-155008 severely inhibited SFN-stimulated *NEAT1* expression in all three cell lines (Fig. 4C). SFN-induced *NEAT1* expression was also inhibited by HS-72 (Fig. S4C). Furthermore, inhibiting Hsp70 activity significantly reduced basal *NEAT1* expression in HeLa cells. The inhibitory effect of VER-155008 on *NEAT1* expression

in HeLa cells was rapid, as a reduction was already detectable after 1 h of incubation with the drug and was maximal after 2 h (Fig. 4D). Of note, the inhibitory effect of VER-155008 seemed to be more pronounced on *NEAT1_2* than *NEAT1* (*NEAT1_1*+*NEAT1_2*), suggesting that Hsp70 has a particularly important role in regulating *NEAT1_2* expression.

## Nucleolar retention of Hsp70 is associated with inhibition of proteotoxic stress-induced paraspeckle formation

Our proteomic data indicate that stress-induced interaction between Hsp70 and NONO is dependent on paraspeckle formation. This suggests that Hsp70 and NONO colocalize during stress in wild-type cells. To investigate this, we employed a live-cell imaging approach to be able to monitor the dynamics of both paraspeckles and Hsp70 in cells subjected to proteotoxic stress. To this end, we generated a HeLa Flp-In cell line where paraspeckles are visualized by fusion of GFP to endogenous SFPQ. We then integrated the *HSPA1B* gene fused to the gene encoding mCherry into the FRT site allowing DOX-inducible expression of mCherry–Hsp70 (Fig. 5A). Immunoblot analyses verified constitutive expression of GFP–SFPQ and DOX-inducible expression of mCherry–Hsp70 (Fig. 5B,C). We treated the GFP–SFPQ/mCherry–Hsp70-expressing HeLa cells with SFN in the absence or presence of VER-155008, and investigated them by live-cell imaging for 6 h. SFN promoted the formation of paraspeckles that started to become visual after 60 min of incubation and remained as distinct dots throughout the incubation time (Fig. 5D; Movie 1). We could, however, not detect colocalization of mCherry–Hsp70 and GFP–SFPQ in paraspeckles. As expected, addition of VER-155008 severely reduced SFN-induced paraspeckle formation (Fig. 5E; Movie 2). Interestingly, but perhaps not surprisingly, incubation of the GFP–SFPQ/mCherry–Hsp70 cell line in the presence of both SFN and VER-155008, led to a distinct localization of mCherry–Hsp70 to structures in the nuclei most likely corresponding to the nucleoli.

We then performed a similar experiment where proteotoxic stress was induced by subjecting the cells to elevated temperature (heat shock, HS). The reason for this was that HS has been previously shown to promote clear and distinct nuclear translocation of Hsp70, and we envisioned that this would make it easier to study a potential colocalization between paraspeckles and Hsp70 (Welch and Feramisco, 1984; Alastalo et al., 2003; Kose et al., 2012; Pelham, 1984; Velazquez and Lindquist, 1984). Moreover, *NEAT1* expression and paraspeckle formation have been shown to be induced by HS (Lellahi et al., 2018). We exposed the GFP–SFPQ/mCherry–Hsp70-expressing cells to HS (43°C, 30 min) in the absence or presence of VER-155008, and left them to recover at 37°C for up to 6 h. The dynamics of GFP–SFPQ and mCherry–Hsp70 were monitored during the whole period by time-lapse imaging (Fig. 6A, upper panel; Movie 3). As demonstrated previously by others, HS induced a rapid nuclear translocation and clear localization of mCherry–Hsp70 to subnuclear structures. (Fig. 6A,B; Movie 3) (Pelham, 1984; Welch and Feramisco, 1984; Alastalo et al., 2003). We verified that these structures indeed corresponded to nucleoli by performing immunofluorescent staining of fixed cells with an antibody recognizing nucleolin, a protein that primarily localizes to the nucleolus (Fig. S5). Nucleolar localization was reversible as mCherry–Hsp70 relocated to the nucleoplasm after ~180 min of recovery, and then started to return to the cytoplasm (Fig. 6A, upper panel, B). In the same experiment, GFP–SFPQ immediately formed small speckles in the nucleus. This preceded its assembly into larger bodies that became clearly visual after 180-240 min of recovery (Fig. 6A, upper panel, C). The formation of the latter

coincided with the kinetics of HS-induced *NEAT1_2* expression (Fig. 6D) and punctuated colocalization of GFP-SFPQ and *NEAT1_2* (Fig. S6), suggesting that these bodies correspond to bona fide paraspeckles. The kinetics of heat stress-induced paraspeckle formation clearly suggest that they primarily assemble during recovery after Hsp70 has relocated from the nucleolus to the nucleoplasm (Fig. 6A, upper panel, B,C). Importantly, HS-induced paraspeckle formation and *NEAT1_2* expression, as well as redistribution of Hsp70 from the nucleoli to the nucleoplasm, were inhibited by VER-155008, demonstrating that all these events are dependent on Hsp70 activity (Fig. 6A, lower panel, B–D; Fig. S5, Movie 4).

## NONO is retained in the nucleolus upon inhibition of Hsp70 activity

A striking observation from inhibiting Hsp70 activity in SFN-treated U2OS cells, was that endogenous NONO relocated into large subnuclear structures (Fig. 4A). As NONO has previously been shown to redistribute to the nucleolus upon cellular stress, we postulated that these structures, at least partially, correspond to nucleoli (Trifault et al., 2024; Moore et al., 2011; Fox et al., 2002; Trifault et al., 2022; Yasuhara et al., 2022). To determine this, we treated U2OS cells with SFN in the absence or presence of VER-155008 and coimmunostained with antibodies specific for NONO and nucleolin. Indeed, whereas NONO displayed diffuse localization in the nucleoplasm and no colocalization with nucleolin in untreated control cells and cells that were treated solely with VER-155008 or SFN, NONO clearly colocalized with nucleolin in ~60% of the cells that were incubated in the presence of both SFN and VER-155008 (Fig. 7A,B). This suggests that inhibition of Hsp70 activity can lead to nucleolar detention of NONO upon SFN treatment. Paraspeckle formation is strictly dependent on the binding of NONO to *NEAT1_2* transcript. It is therefore easy to envision that nucleolar retainment of NONO will affect paraspeckle assembly. To study this more closely, we integrated *NONO* fused to the mCherry gene into the DOX-inducible FRT site of the GFP–SFPQ-expressing HeLa Flp-In cell line, allowing real-time monitoring of mCherry–NONO dynamics (Fig. S7). We incubated GFP–SFPQ/mCherry–NONO-expressing cells in the absence or presence of VER-155008 and subjected them to HS and recovery. Interestingly, whereas GFP–SFPQ remained in the nucleoplasm, mCherry–NONO was also detected in the nucleoli immediately after HS and for a period of ~120 min during recovery before returning to the nucleoplasm (Fig. 7C, upper panel, D; Table S2, Movie 5). The nucleolar localization of mCherry–NONO became more pronounced upon VER-155008 treatment and the protein failed to recycle back to the nucleoplasm during recovery from HS (Fig. 7C, lower panel, D; Fig. S8, Table S2, Movie 6). Trapping of NONO in the nucleoli might, at least partially, explain why paraspeckles fail to form upon inhibition of Hsp70 activity during proteotoxic stress.

## DISCUSSION

Members of the Hsp70 chaperone family have emerged as crucial regulators of biomolecular condensates. A growing number of studies have reported key roles of Hsp70 and their cochaperones in modulating the biophysical properties of stress granules and TDP-43-containing nuclear condensates to ensure their timely disassembly and prevent them from undergoing liquid-to-solid phase transition (Ganassi et al., 2016; Mateju et al., 2017; Mazroui et al., 2007; Li et al., 2022; Akaree et al., 2025; Gu et al., 2021; Yu et al., 2021; Agnihotri et al., 2025). Here, we add another layer of knowledge to Hsp70-mediated regulation of RNA condensates by showing that

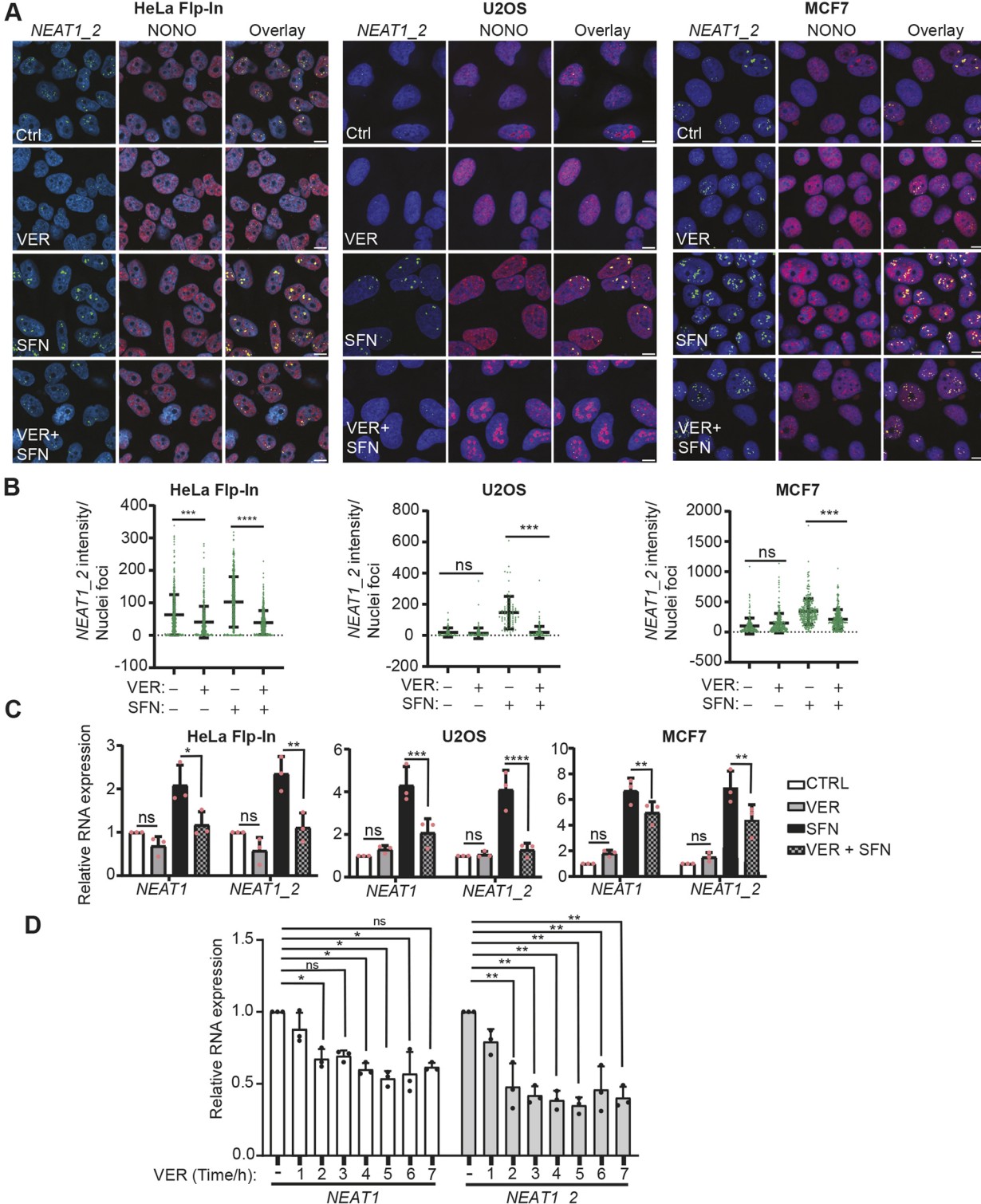

**Fig. 4. SFN-induced paraspeckle formation and *NEAT1* expression are dependent on Hsp70 activity.** (A) HeLa Flp-In, U2OS and MCF7 cells were left untreated or treated with 20 µM SFN for 6 h in the absence or presence of 25 µM (HeLa Flp-In and U2OS) or 50 µM (MCF7) VER-155008 (VER). Paraspeckles were visualized by *NEAT1_2* (green)- and NONO (red)-specific co-immuno-FISH analyses. The nuclei are visualized by DAPI (blue). (B) The intensity of the *NEAT1_2* signals was quantified using the ImageJ software in at least 200 cells per cell line treated as described above. The quantification was done across three independent experiments. ***$P \leq 0.001$; ****$P \leq 0.0001$; ns, not significant (one-way ANOVA with Tukey's multiple comparison test). (C) Hsp70 inhibition reduces SFN-induced *NEAT1* expression. Cells were treated as described in A and RNA was isolated. Relative expression of *NEAT1* (*NEAT1_1*+*NEAT1_2*) and *NEAT1_2* in treated versus untreated cells was determined by RT-qPCR analyses. The mean±s.d. of three replicates is presented. *$P \leq 0.05$; **$P \leq 0.01$; ***$P \leq 0.001$; ns, not significant (two-way ANOVA with Tukey's multiple comparison test). (D) VER-155008 reduces baseline *NEAT1* expression in HeLa Flp-In cells. HeLa cells were treated with VER-155008 for the indicated time periods. *NEAT1* (*NEAT1_1*+*NEAT1_2*) and *NEAT1_2* expression was determined by RT-qPCR. The mean±s.d. of three replicates is presented. *$P \leq 0.05$; **$P \leq 0.01$; ns, not significant (two-way ANOVA with Dunnett's multiple comparison test). All error bars are mean±s.d. Scale bars: 10 µm.

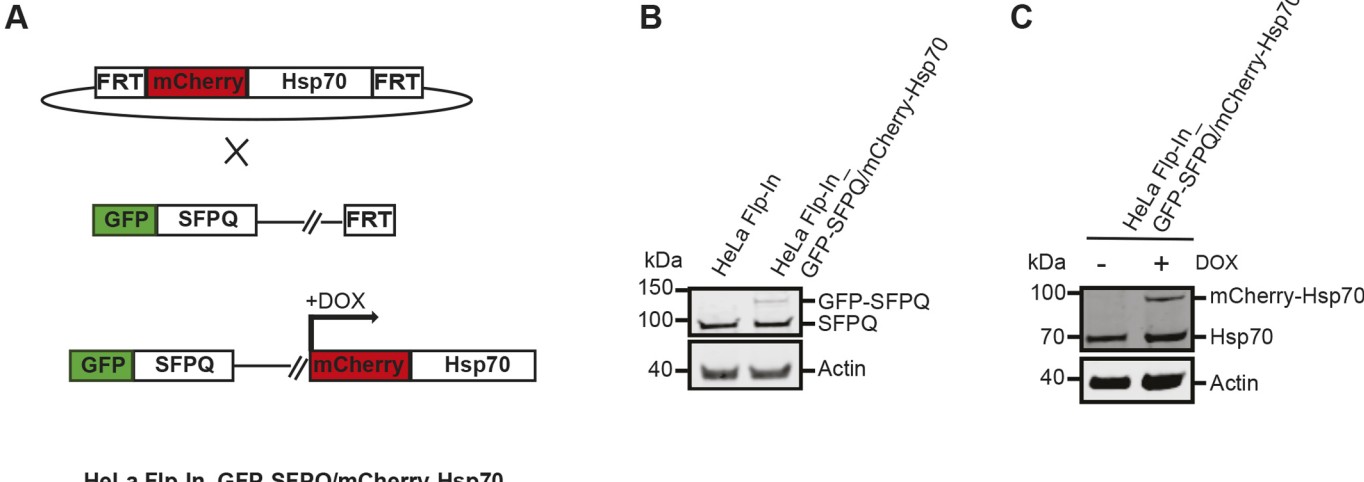

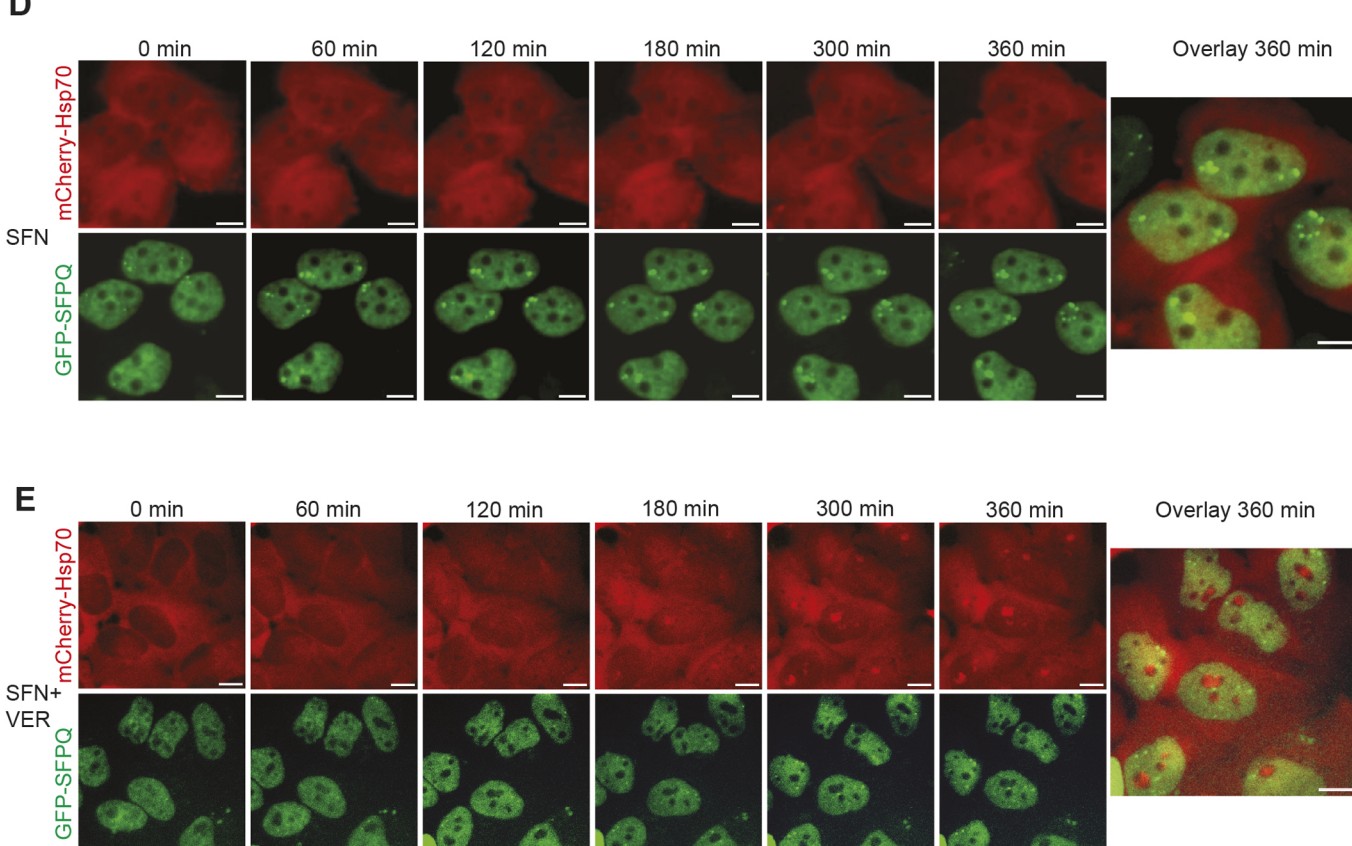

**Fig. 5. Hsp70 does not localize to SFN-induced paraspeckles.** (A) Schematic illustration of the generation of the HeLa Flp-In_GFP-SFPQ/mCherry-Hsp70 cell line. GFP was N-terminally fused to endogenous SFPQ in HeLa Flp-In cells. A construct encoding mCherry–Hsp70 was integrated into the FRT site of the HeLa Flp-In_GFP-SFPQ cell line, which allows doxocyline (DOX)-inducible expression of mCherry–Hsp70. (B) Expression of GFP–SFPQ in HeLa Flp-In_GFP-SFPQ/mCherry-Hsp70 cells was verified by immunoblotting using an anti-SFPQ antibody. Equal loading was verified by re-probing the membrane with an anti-actin antibody. Note that the cell line still expresses untagged endogenous SFPQ, indicating that *GFP* is inserted in front of only one of the *SFPQ* alleles. (C) The HeLa Flp-In_GFP-SFPQ/mCherry-Hsp70 cell line expresses mCherry–Hsp70 in a DOX-inducible manner. mCherry–Hsp70 and endogenous Hsp70 are visualized by immunoblot analyses using an anti-Hsp70 antibody. Equal loading was verified by re-probing the membrane with an anti-actin antibody. (D) Timelapse imaging analyses of the HeLa Flp-In_GFP-SFPQ/mCherry-Hsp70 cell line incubated in the presence of SFN for 6 h. *Z*-stack images were acquired every 30 min by widefield microscopy. Cropped representative images are shown. (E) VER-155008 (VER) interferes with assembly of GFP–SFPQ into paraspeckles and leads to nucleolar retention of mCherry–Hsp70 in SFN-treated cells. Time lapse imaging analyses of the HeLa Flp-In_GFP-SFPQ/mCherry-Hsp70 cell line incubated in the presence of VER-155008 and SFN for 6 h. Images were obtained as described in D. Images in B–E are representative of at least three repeats. Scale bars: 10 μm.

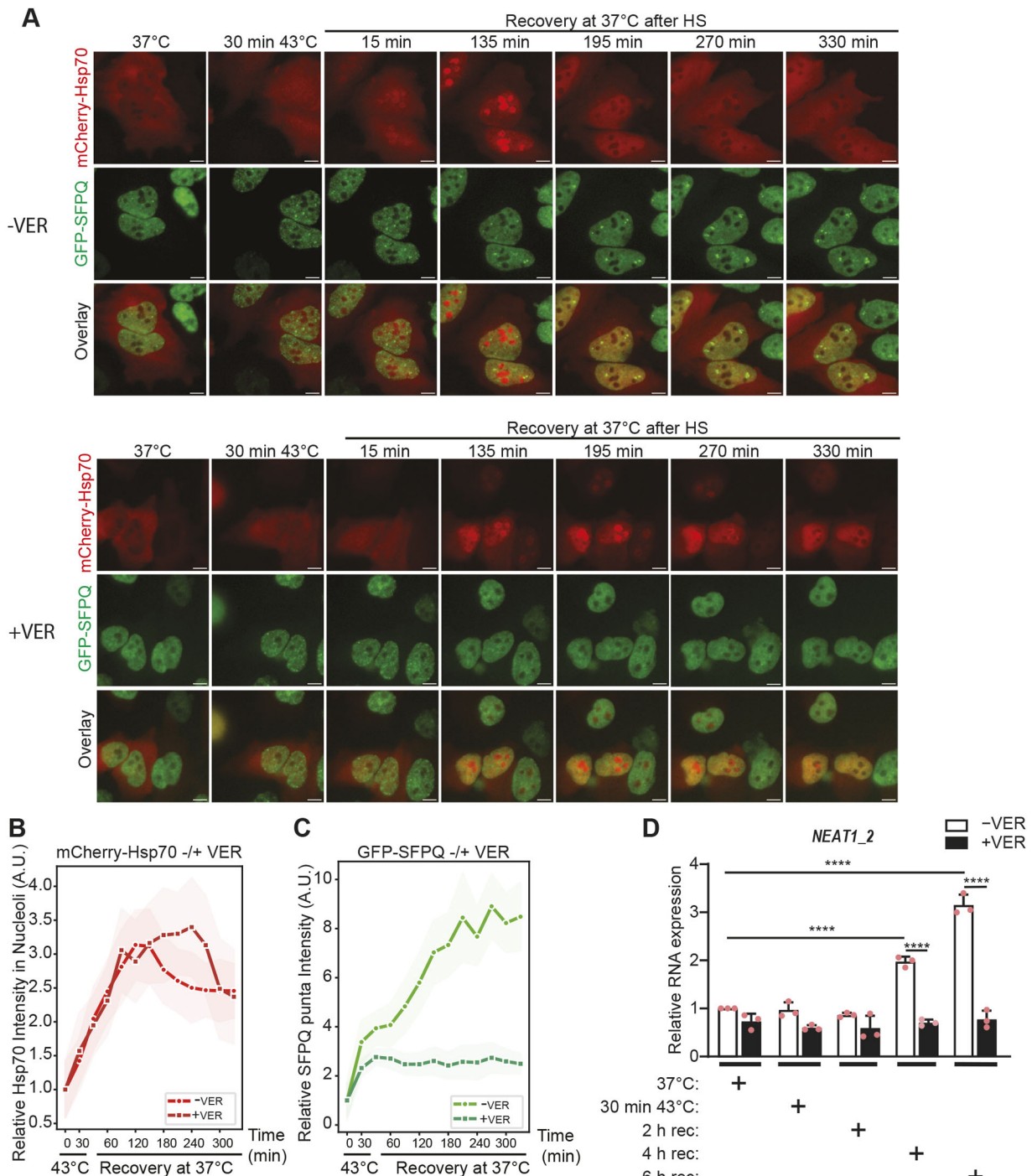

**Fig. 6. Paraspeckle assembly during post-heat shock recovery follows nucleolar exit of Hsp70.** (A) Timelapse imaging of HeLa Flp-In_GFP-SFPQ/ mCherry-Hsp70 cells that were exposed to HS (43°C, 30 min) in the absence (upper panel) or presence (lower panel) of VER-155008 (VER), and left to recover at 37°C. Z-stack images were acquired every 30 min by widefield microscopy. Cropped representative images are shown. (B) Kinetics of nucleolar translocation of mCherry–Hsp70 during HS and recovery in cells that were either left untreated or treated with VER-155008. Nucleolar localization of mCherry–Hsp70 was quantified in at least 200 cells per time point across three independent experiments using the CellProfiler software. (C) Kinetics of the assembly of GFP–SFPQ in paraspeckles during recovery from HS. The intensity of punctuated GFP-SFPQ signals was quantified in the same cells as in B using the CellProfiler software. (D) The assembly of GFP–SFPQ into paraspeckles during recovery coincides with Hsp70-dependent elevation of *NEAT1_2* expression. HeLa Flp-In_GFP-SFPQ/mCherry-Hsp70 cells were treated as described in A and RNA was isolated. Relative *NEAT1_2* expression compared to untreated control cells was determined by RT-qPCR. Results are mean±s.d. (*n*=3). ****$P \leq 0.0001$ (one-way ANOVA with Tukey's multiple comparison test). A.U., arbitrary units. Scale bars: 10 µm.

Hsp70 activity is required for stress-induced paraspeckle formation. We identified Hsp70 among proteins that became enriched in complexes with the essential paraspeckle protein NONO in response

to two different stressors. The stress-induced enrichment of Hsp70 in NONO complexes was not observed in paraspeckle-deficient *NEAT1*-knockout cells, suggesting that *NEAT1* is required for

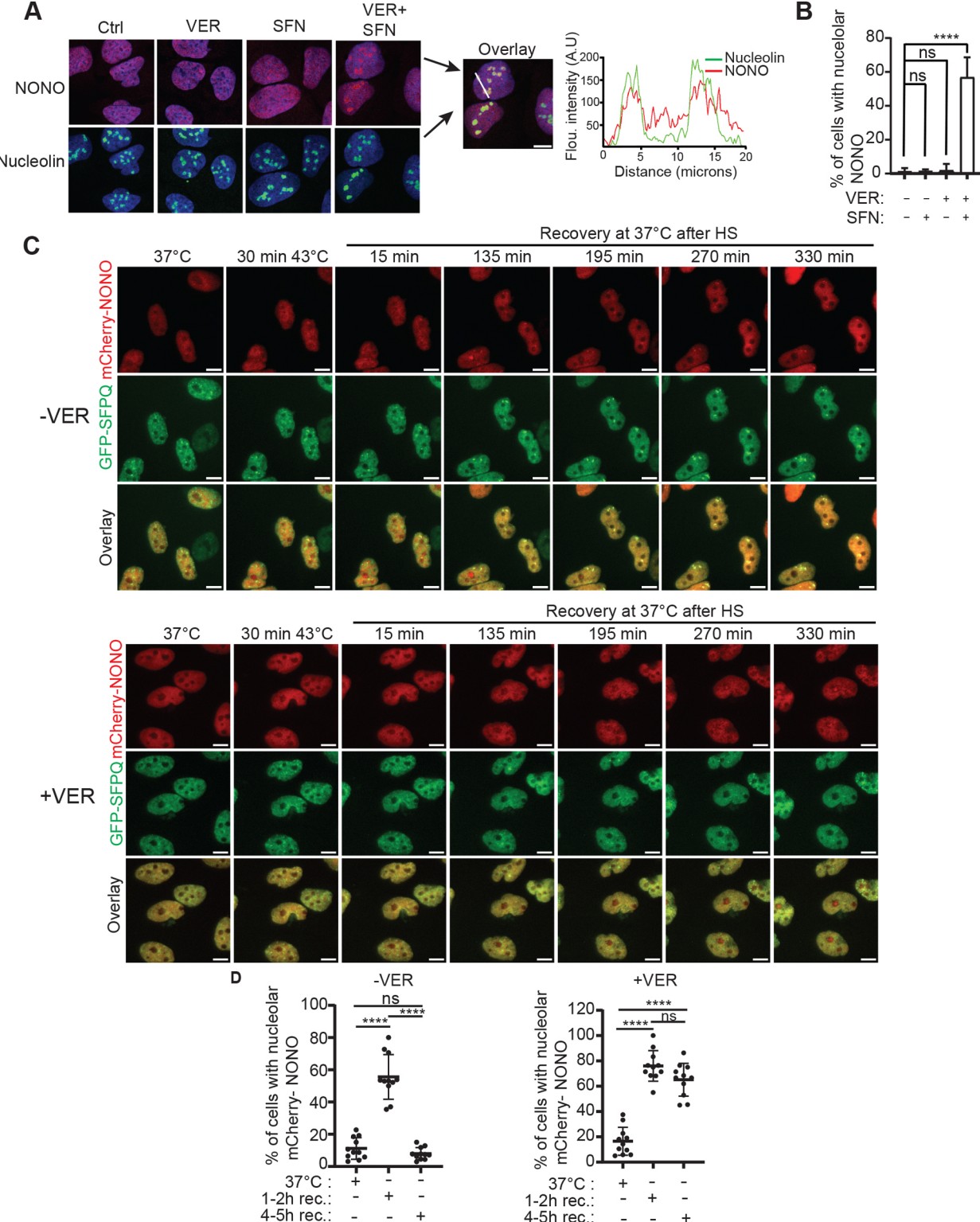

**Fig. 7. NONO is retained in the nucleolus upon inhibition of Hsp70 activity.** (A) U2OS cells were left untreated or treated with SFN (20 µM), VER-155008 (VER, 25 µM) or a combination of both reagents. Cells were fixed and coimmunostained with antibodies recognizing NONO and nucleolin. (B) The percentage of cells displaying colocalization of NONO and nucleolin in cells treated as described in A. Quantification was done in at least 177 cells per condition from microscopy images from three independent experiments. ****$P\leq$0.0001; ns, not significant (one-way ANOVA with Dunnett's multiple comparison test). (C) Timelapse imaging of HeLa Flp-In_GFP-SFPQ/mCherry-NONO cells that were exposed to HS (43°C, 30 min) in the absence (upper panel) or presence (lower panel) of VER-155008 and left to recover at 37°C. Z-stack images were acquired every 30 min by widefield microscopy. Cropped representative images are shown. (D) Quantification of the experiments shown in C. The percentage of cells displaying nucleolar localization of mCherry–NONO before and during early (1–2 h) and late (4–5 h) recovery from HS are shown. Each dot represents quantification of cells within one image taken at 40× objective magnification with total numbers of analyzed cells ranging from 7 to 34 between the different images (Table S2). ****$P\leq$0.0001; ns, not significant (one-way ANOVA with a Tukey's multiple comparisons test). All error bars are mean±s.d. A.U., arbitrary units. Scale bars: 10 µm.

bringing Hsp70 into the proximity of NONO. A recent report has shown that *NEAT1* is essential for recruiting Hsp70 into stress-induced TDP-43 nuclear condensates, which is necessary for maintaining their fluidity and avoiding the formation of cytotoxic aggregates (Agnihotri et al., 2025). This made us hypothesize that Hsp70 is also recruited to stress-induced paraspeckles in a *NEAT1*-dependent manner. However, recombinant Hsp70 did not localize to SFN-induced paraspeckles nor to paraspeckles that formed during recovery in cells that had been exposed to HS. Instead, both NONO and Hsp70 translocated to the nucleolus upon proteotoxic stress. Stress-promoted nucleolar translocation of NONO was further underscored by our proteomic data showing that both SFN- and starvation-induced NONO interactomes were enriched in proteins that normally reside in the nucleolus. This is consistent with the pioneer work by Archa Fox and coworkers demonstrating that NONO indeed shuttles between paraspeckles and nucleolar structures (Fox et al., 2005, 2002). Furthermore, NONO has been found to accumulate in the nucleolus in response to a range of cellular stressors (see below) (Trifault et al., 2022, 2024; Fox et al., 2005, 2002; Shav-Tal et al., 2005; Andersen et al., 2002; Yasuhara et al., 2022).

The heat stress-induced relocation of both Hsp70 and NONO to the nucleolus was transient, as both proteins returned to the nucleoplasm during recovery. Importantly, the exit of both proteins from the nucleolus was strictly dependent on Hsp70 activity. It is becoming increasingly clear that the nucleolus, in addition to being the site of ribosome biogenesis, plays a key role in nuclear protein quality control, and nucleolar translocation of Hsp70 in response to heat shock is well-established (Amer-Sarsour and Ashkenazi, 2019; Frottin et al., 2019; Nollen et al., 2001; Welch and Feramisco, 1984; Alastalo et al., 2003; Kose et al., 2012; Pelham, 1984). Many nuclear RNA-binding proteins contain low-complexity domains, which make them prone to undergoing aggregation in response to stress (Amer-Sarsour and Ashkenazi, 2019; Frottin et al., 2019). Recently, it has been demonstrated that HS-induced aggregation of nuclear proteins is mitigated by transfer of unfolded proteins and Hsp70 to the granular component (GC) of the nucleolus (Frottin et al., 2019). Here, the misfolded proteins are stored in a state that prevents them from forming amyloid-like aggregates until Hsp70-assisted refolding promotes their redistribution back to the nucleoplasm during recovery. Of relevance to our study, NONO was indeed identified among ~200 proteins that became enriched in complexes with the GC protein nucleophosmin during HS (Frottin et al., 2019).

By conducting timelapse analyses of GFP–SFPQ/mCherry–Hsp70- and GFP–SFPQ/mCherry–NONO-expressing cell lines, we found that HS-induced paraspeckles formed during recovery after both NONO and Hsp70 had left the nucleoli. Their formation coincided with elevated *NEAT1* expression. Importantly, both *NEAT1* expression and paraspeckle formation were severely reduced upon Hsp70 inhibition. As NONO is strictly required for paraspeckle assembly and *NEAT1* expression, it is possible that detention of NONO in the nucleoli upon Hsp70 inhibition directly impairs paraspeckle formation during recovery (Naganuma et al., 2012). This is similar to what has been reported for a range of epigenetic modifiers, including several subunits of polycomb repressive complex 1 (PRC1) and PRC2, which reversibly translocate to the nucleoli upon HS (Azkanaz et al., 2019). Their redistribution back to the nucleoplasm during recovery to resume chromatin modification is dependent on Hsp70 activity. As such, the nucleolus appears to be an essential protein quality control site that ensures proper maintenance of the epigenome following proteotoxic stress. Keeping with this notion, transfer of NONO to the nucleoli during heat stress might be a

prerequisite for paraspeckle formation during recovery. Importantly, NONO has been demonstrated to naturally translocate to nucleolar structures in healthy cells in a cell cycle-dependent manner (Fox et al., 2005). Paraspeckles are absent in newly formed nuclei in telophase, which coincides with transient localization of NONO at perinucleolar caps. In early G1 phase, NONO is redistributed back to the nucleoplasm where it displays a diffuse localization for a period before being assembled into paraspeckles. Future studies should be undertaken to determine whether perinucleolar localization of NONO and chaperone activity at the end of mitosis are required for *de novo* paraspeckle formation during G1 phase. Notably, we show that inhibition of Hsp70 family members by VER-155008 quickly reduces baseline *NEAT1_2* expression and paraspeckle formation in normally cycling HeLa cells.

NONO is a multifunctional nuclear protein that plays many roles in transcriptional and post-transcriptional regulation of gene expression, as well as in regulation of DNA repair. We present evidence that proteotoxic stress, and possibly starvation, promote relocation of NONO to the nucleolus. Importantly, other stressors including transcription inhibition, DNA damage, nucleolar stress and cold shock also promote nucleolar accumulation of NONO (Fox et al., 2005, 2002; Shav-Tal et al., 2005; Andersen et al., 2002; Trifault et al., 2022, 2024; Yasuhara et al., 2022). Moreover, NONO naturally accumulates at perinucleolar caps at the end of mitosis (Fox et al., 2005). Given the spectrum of conditions that lead to nucleolar detention of NONO, it is not unlikely that the biological significance of this reaches beyond simply safeguarding its protein quality. For instance, nucleolar translocation of NONO in response to DNA damage serves to tether a specific subset of pre-mRNAs to the nucleoli in response to DNA damage, which facilitates the repair of double-strand DNA breaks at their corresponding genomic loci by preventing aberrant R-loop formation (Trifault et al., 2024). NONO and SFPQ have also been shown to form rRNA-containing nucleolar condensates upon transcription inhibition referred to as 'condensates induced by transcription inhibition' (CITIs), which tether SFPQ-bound active chromatin to the nucleoli (Yasuhara et al., 2022). Although the physiological significance of formation of CITIs remains obscure, chromatin that localizes to these structures is prone to forming aberrant genome fusions, illustrating the existence of a link between transcription inhibition and genomic instability. Whether NONO also plays an active role in nucleolar processes, such as ribosome biogenesis, remains to be determined.

In our proteomic data, we identified Hsp70 as a stress-induced partner of NONO in wild-type but not in *NEAT1*-knockout cells. The underlying mechanism for this remains to be resolved. We have not demonstrated that *NEAT1* is required for a direct interaction between Hsp70 and NONO. Moreover, we have tried to pull down both endogenous Hsp70 and NONO and examined the co-immunoprecipitation of the other protein, using altogether four different antibodies. However, these experiments have not been successful, which might be due to antibody-related technical issues. Our proteomic data are, however, supported by the fact that although *NEAT1* is not detected in nucleolar CITIs, the recruitment of NONO and SFPQ to these structures is severely reduced in *NEAT1*-depleted cells (Yasuhara et al., 2022). Generally, NONO and SFPQ form heterodimers that bind to the middle segment of *NEAT1_2* to promote paraspeckle assembly, and the proteins frequently operate as heterodimers in other cellular processes (Huang et al., 2018). In line with this, both proteins have been reported to accumulate in the nucleolus during stress (Yasuhara et al., 2022; Trifault et al., 2024). It is perhaps therefore surprising that we did not detect nucleolar translocation of endogenous SFPQ

Journal of Cell Science

fused to GFP upon proteotoxic stress. One possible explanation could be that nucleolar localization of SFPQ is cell line-dependent and that GFP–SFPQ is relatively thermostable in HeLa cells.

An important limitation of our study of stress-specific NONO interactomes is the small number of proteins that became enriched in NONO complexes in response to SFN in wild-type cells compared to in *NEAT1*-knockout cells. At first glance, it is tempting to envision that the presence of *NEAT1* and paraspeckles restrains the number of proteins that become engaged in NONO complexes and that the diffuse localization of NONO in the nucleoplasm of *NEAT1*-deficient cells makes it more promiscuous in forming complexes with other proteins. However, we observed that SFN consistently led to a reduction in the expression of the TurboID–V5–NONO fusion protein in four independent experiments. SFN has indeed been shown to induce the protein quality control system through the heat-shock response, which might recognize and degrade the ectopically expressed artificial fusion protein. Intriguingly, the SFN-mediated reduction of TurboID–V5–NONO expression in *NEAT1*-knockout cells was modest. We therefore speculate that *NEAT1* and paraspeckles play an active role in nuclear protein quality control. This is currently a focus of ongoing research in our lab. A link between *NEAT1* and Hsp70 in proteostasis and neurodegeneration was indeed recently reported by Kawakami et al., who demonstrated that cervical spinal cords from *Neat1*-knockout mice displayed reduced *Hspa1A*/*Hspa1B* expression and Hsp70 protein levels compared to that in wild-type mice (Kawakami et al., 2025). They suggested that the reduction in chaperone levels contributed to increased aggregation of misfolded superoxide dismutase 1 (SOD1) and exacerbated neurodegeneration in *Neat1*-deficient mice. We did not see any reduction in *HSPA1A* expression or Hsp70 levels in *NEAT1*-depleted MCF7 cells compared to the mother cell line, indicating that this mechanism is cell type specific.

We show that proteotoxic stress-induced paraspeckle formation is strictly dependent on Hsp70 activity and that their formation during recovery from heat shock follows nucleolar exit of both Hsp70 and the essential core paraspeckle protein NONO. Paraspeckles appear to be global sensors of cellular stress (McCluggage and Fox, 2021). As such, it is easy to envision that the transient assembly of certain proteins and RNA molecules into these highly dynamic structures contributes to maintaining homeostasis during stress. We find that following heat shock, paraspeckles form primarily during the recovery phase. Previous results from our laboratory have suggested that *NEAT1*-depletion prolonged the heat-shock response (Lellahi et al., 2018). Thus, it is likely that paraspeckles facilitate efficient resumption of normal cellular activities after stress. As increasing evidence clearly suggests that paraspeckles regulate different aspects of RNA metabolism to modify gene expression, it is tempting to speculate that they contribute to reestablishing gene expression patterns following stress. This should be a topic of future research.

This study provides two stress-enriched NONO interactomes identified in wild-type and paraspeckle-deficient *NEAT1*-knockout cells. These data complement previously reported NONO interactomes and will contribute to increased understanding of the functions of NONO and paraspeckles in cellular stress responses (Trifault et al., 2022, 2024; Dyakov et al., 2024 preprint).

## MATERIALS AND METHODS
### Cell lines, treatments, and transfection
MCF7 (ATCC® HTB-22™), HeLa (ATCC® CCL-2™) and U2OS (ATCC® HTB-96™) cells were purchased from the America Type Culture Collection (ATCC). Flp-In T-REx HeLa cells were a kind gift from Terje Johansen, UiT

the Arctic University of Tromsø, Norway. MCF7 and HeLa cells were cultured in low-glucose Dulbecco's modified Eagle's medium (DMEM; Sigma-Aldrich, D6429) supplemented with 10% fetal bovine serum (FBS; Sigma-Aldrich, F7524) and 1% penicillin-streptomycin (Sigma-Aldrich, P4333). MCF7 cells were additionally supplemented with 1 µg/ml insulin (Sigma-Aldrich, 19278). U2OS cells were cultivated in high-glucose DMEM (Sigma-Aldrich, D5796) supplemented with 10% FBS and 1% penicillin-streptomycin. All cells were cultured at 37°C in humidified condition containing 5% $CO_2$ and routinely tested for mycoplasma contamination. The expression of TurboID–V5–NONO, mCherry–Hsp70, and mCherry–NONO was induced by adding 1 µg/ml doxycycline (DOX, Sigma-Aldrich, D9891) for 24 h unless otherwise stated. In preparation of biotinylated proteins, biotin (Sigma-Aldrich, 4501) was added at 100 µM for 30 min before cell lysis. Sulforaphane (SFN, Sigma-Aldrich, S4441) was added to the cells at a final concentration of 20 µM for 6 h. For starvation, cells were washed thoroughly with Hank's balanced salt solution (HBSS, Thermo Fisher Scientific, 14025092) before being left in the solution for 6 h. For heat shock (HS), cells were subjected to 43°C for 30 min and then returned to 37°C for recovery. VER-155008 (MedChemExpress, HY-10941) and HS-72 (Sigma-Aldrich, SML1325) were added to cells at a final concentration of 25 µM. The inhibitors were added to the cells 1 h prior to SFN or HS and left on the cells throughout the experiments. Cells were transfected using Lipofectamine™ 2000 (Thermo Fisher Scientific, 11668027) according to the instructions from the manufacturer.

### Plasmids and generation of stable cell lines
CRISPR/Cas9 genome editing was used to generate MCF7 *NEAT1*-knockout cells (MCF7 N1_KO). *NEAT1*-targeting guide RNAs were designed using the CHOPCHOP online tool (https://chopchop.cbu.uib.no). The sense and antisense oligonucleotides were annealed and phosphorylated, and then cloned into the BbsI restriction sites of pSpCas9(BB)-2A-Puro (PX459) [Addgene #62988, deposited by from Feng Zhang (Ran et al., 2013)]. Following transfection, cells were grown in the presence of 1 µg/ml puromycin (Sigma-Aldrich, P8833) for 72 h. Puromycin-resistant cells were single-cell sorted by fluorescence-activated cell sorting (FACS) into 96-well plates. Clones were expanded and screened for *NEAT1* deletion by DNA sequencing of an amplified PCR product comprising the targeted region. A plasmid expressing TurboID–V5–NONO from a DOX-inducible promoter was generated by amplifying the NONO cDNA from GFP–NONO [a kind gift from Tatyana Shelkovnikova, University of Sheffield, UK (An et al., 2019)] by PCR using primers containing Gateway compatible attB sequences. The PCR product was cloned into the pDONR221 vector (Thermo Fisher Scientific) generating pENTR-NONO. A gateway-compatible lentiviral vector encoding V5-tagged TurboID was made by subcloning TurboID-V5 from C1(1-29)-TurboID-V5_pLX304 [Addgene #107175, deposited by Alice Ting (Branon et al., 2018)] into pCW57.1 (Addgene #41393, deposited by David Root), generating pCW57.1_TurboID_V5 (a kind gift from Mireia Nager and Terje Johansen, UiT the Arctic University of Norway, Norway). NONO was then inserted into pCW57.1_TurboID_V5 by Gateway recombination cloning. Lentiviral transduction was used to generate MCF7 and MCF7 N1_KO cell lines with DOX-inducible expression TurboID-V5-NONO. In brief, HEK 293T cells were transfected with the pCW57_TurboID-V5_NONO plasmid together with psPAX2 and pMD2.G for lentiviral packaging (Addgene #12260 and Addgene #12259, deposited by Didier Trono). Media containing viral particles were harvested at 48 h and 72 h post-transfection. MCF7 and MCF7 N1_KO cells were transduced with the lentiviral particles in the presence of 8 µg/ml hexadimethrine bromide (polybrene; Sigma-Aldrich, H9268) and puromycin-resistant cell populations were selected. Flp-In T-REx HeLa cells with endogenous expression of GFP–SFPQ (HeLa Flp-In_GFP-SFPQ) was made by CRISPR/Cas9 knock-in as described previously (Li et al., 2017) using pX330-GFP-SFPQ and pUC57-GFP-SFPQ-RT (Addgene #97084 and Addgene #97090, deposited by Archa Fox). The HeLa Flp-In_GFP-SFPQ cell lines with DOX-inducible expression of mCherry–Hsp70 or mCherry–NONO were made by the Flp-In™ system (Invitrogen™). The *HSPA1A* cDNA was amplified from EGFP–Hsp70 [Addgene #15215, deposited by Lois Greene (Zeng et al., 2004)] by PCR using primers containing attB sites and cloned into pDONR221 generating pENTR-Hsp70. Hsp70 and NONO were subcloned from the pENTR-Hsp70

and pENTR-NONO into pDest-mCherry-Flp-In-FRT/TO (kind gift from Terje Johansen, UiT The Arctic University of Norway, Norway) by gateway recombination cloning. The resulting pDest-mCherry-Hsp70-FRT/TO and pDest-mCherry-NONO-FRT/TO plasmids were transfected into HeLa Flp-In_GFP-SFPQ cells together with pOG44 (Thermo Fisher Scientific, V600520). HeLa Flp-In_GFP-SFPQ/mCherry-Hsp70 and HeLa Flp-In_GFP-SFPQ/mCherry-NONO cells lines were selected by cultivation in medium containing 200 µg/ml hygromycin B (Calbiochem, 400051). The sequences of the guide RNAs and PCR primers are provided in Table S3.

### RNA isolation, cDNA synthesis and RT-qPCR
RNA was isolated from cells cultivated in 24-well plates using the GenElute Mammalian Total RNA Miniprep Kit (Sigma-Aldrich, RTN350). Cell lysates were heated for 10 min at 55°C before RNA was isolated according to the protocol provided by the manufacturer. cDNA was synthesized from 200 ng RNA using Superscript™ IV Reverse Transcriptase (Thermo Fisher Scientific) and 2.5 µM random hexamer primer (Thermo Fisher Scientific, P/N100026484). Quantitative PCR was run on a LightCycler 96 (Roche Life Science) using FastStart Essential DNA green Master (SYBR green detection) (Roche, 06402712001) and a 0.25 µM forward and reverse primer. Steps for the thermal cycle are 95°C for 10 min and 40 cycles of 95°C for 10 s, 60°C for 10 s and 72°C for 10 s. Experiments were done in triplicates and GAPDH was used as a reference gene for normalization. The $\Delta\Delta$Cq method was used for fold change calculations. All primer sequences are provided in Table S3.

### RNA-fluorescence in situ hybridization, immunofluorescence staining and confocal imaging
Stellaris® NEAT1_2 probes (Human NEAT1_m with Quasar® 670 Dye, VSMF-2251-5) were purchased from Biosearch technologies. Cell preparation, hybridization and mounting of coverslips were performed according to Stellaris® RNA FISH probes protocol. Briefly, cells were grown on coverslips in 24-well dishes. They were washed with RNase-free 1× PBS (Thermo Fisher Scientific, AM9625), fixed with 4% paraformaldehyde (PFA) for 15 min on ice and then permeabilized overnight with ice-cold 75% ethanol. Hybridization was done overnight at 37°C in a humidifying chamber with 125 nM NEAT1_2 probes in hybridization buffer [10 mg/ml dextran sulphate, 10% formamide in 2× SSC buffer (ThermoFisher Scientific, 15557044)]. For co-immuno-FISH experiments, cells were fixed and permeabilized as described above and then incubated overnight at 37°C with 50% formamide in 2× SSC buffer. Cells were subsequently blocked with 1% RNase-free bovine serum albumin (BSA; Thermo Fisher Scientific, AM2616) for 30 min and incubated with primary antibody for 1 h. Cells were washed six times with RNase-free 1× PBS and incubated with Alexa Fluor®488-conjugated secondary antibodies (Thermo Fisher Scientific) together with DAPI for 1 h. Cells were washed six times in RNase-free 1× PBS and subjected to hybridization as described above. The coverslips were washed twice for 30 min with wash buffer (10% formamide in 2× SSC) at 37°C and then briefly rinsed in 2× SSC buffer before being mounted on microscope slides using Vectashield Vibrance® Antifade Mounting Medium (Vector Laboratories, H-1700). For conventional immunofluorescence staining, cells on coverslips were fixed in 4% freshly made PFA at room temperature (RT) and permeabilized in 0.2% Triton X-100 for 15 min. The coverslips were blocked in 5% BSA for 1 h and incubated in the presence of primary antibodies for 1 h at RT. Following intensive washing in 1× PBS, the cells were incubated with Alexa Fluor®488- or Alexa Fluor®647-conjugated secondary antibodies for 1 h, and then with DAPI for 5 min. The coverslips were washed in 1× PBS and mounted as described above. For all experiments, Z-stack images were acquired with 40× and 63× magnification objectives using a Zeiss LSM 800 confocal microscope (Carl Zeiss Microscopy GmbH, Jena, Germany). Similar image acquisition settings were applied to all samples within an experiment. All images were processed using the ZEN Blue (ZEISS, Version 3.7.97.03000) software. NEAT1_2 signal quantifications were done using ImageJ software (FIJI). Generally, more than 250 cells in each group of treatment were analyzed by the ImageJ software. The mean intensity or gray value of NEAT1_2 foci per nuclei upon treatment were compared to their respective controls. Primary and secondary antibodies are listed in Table S4.

### Immunoblotting and immunoprecipitation
Protein expression was determined in whole-cell extracts (WCEs) or nuclear extracts (NEs). WCE were made by lysing the cells in 1× SDS sample buffer (67.5 mM Tris-HCl pH 6.8, 2% SDS and 10% glycerol). For NEs, cells were washed three times in ice-cold 1× PBS and lysed in ice-cold 0.1% NP-40 in PBS supplemented with protease inhibitors (Halt™ protease inhibitor cocktail, Thermo Fisher Scientific, 78430). Following incubation on ice for 10 min, the lysates were centrifuged at 10,000 $g$ for 10 s and the supernatant was discarded. The pellet was resuspended in 0.1% NP-40 and the previous step was repeated one more time. The pellet containing the nuclei was lysed in RIPA buffer (50 mM Tris-HCl pH 7.4, 150 mM sodium chloride, 1.0% NP-40, 0.5% sodium deoxycholate and 0.1% SDS) supplemented with protease inhibitors and incubated on ice for 10 min. The lysed nuclear fraction was cleared by centrifugation at 13,000 $g$ for 10 min at 4°C. Proteins were resolved by SDS-PAGE and transferred onto a nitrocellulose membrane. To ensure equal loading of proteins, membranes were re-probed with anti-actin (WCE) or anti-Lamin B1 (NE) antibodies. The blots were developed with IRDye®-conjugated secondary antibodies (LI-COR Biosciences) at a 1:10,000 dilution and proteins were detected with the odyssey® CLx Infrared Imaging System. For immunoprecipitation, cells were lysed in RIPA buffer supplemented with protease inhibitors and left on ice for 30 min. The cell lysates were centrifuged at 10,000 $g$ for 10 min at 4°C. GFP-Trap® Agarose beads (Chromotek, gta) were added to the cleared lysate and left on a rotating wheel at 4°C for 1 h. The beads were washed three times in co-IP wash buffer (10 mM Tris-HCl pH 7.5, 150 mM NaCl, 0.05% NP-40, 0.5 mM EDTA). Bound proteins were eluted from the beads by boiling in a 2× SDS sample loading buffer [0.125 M Tris-HCl pH 6.8, 4% (w/v) SDS, 20% (v/v) glycerol and 0.02% Bromophenol Blue]. Primary and secondary antibodies are listed in Table S4. Uncropped images of blots from this paper are presented in Fig. S9.

### Live-cell imaging
Cells were seeded in glass-bottom 24-well plates and imaged at ~70% confluency using a Celldiscoverer 7 automated widefield microscope (ZEISS Microscopy). Z-stack images were acquired with a 40× magnification objective every 15 min across multiple regions of each well, using the built-in autofocus functionality of the system. For heat-shock experiments, the incubator temperature was raised to 43°C and subsequently returned to 37°C, as described above. The estimated lag time for temperature changes was 10–15 min. Image sequences were processed using ImageJ. The nucleolar intensity of mCherry–Hsp70 and punctuated GFP–SFPQ intensity were quantified using CellProfiler image analysis software (version 4.2.8, Broad Institute, USA). Briefly, nuclei were identified as primary objects, and their masks were inverted to define the nucleolar regions, where intensity measurements for mCherry–Hsp70 were performed. GFP–SFPQ signal intensity was quantified by defining the nuclei as primary objects, thresholding for the diffused background signal and measuring integrated intensity within these objects. The resulting data were post-processed in Python, where objects with a lifetime of more than 15 time points across the image sequences were retained. The data were subsequently visualized using Python's Seaborn package. The percentage of cells displaying nucleolar localization of mCherry–NONO at indicated time points during a heat-shock experiment was quantified manually using images extracted from time lapse videos. This was due to inherent cell-to-cell heterogeneity in mCherry–NONO expression levels within the HeLa Flp-In_GFP-SFPQ/mCherry-NONO cell line, and reduced fitness of many cells during the heat shock experiment. Only cells that displayed a healthy morphology for up to 5 h of recovery after heat shock were tracked and included in the calculations. This excluded cells that detached from the plates or developed a multinucleated phenotype during the experiment.

### Preparation of biotinylated protein samples, TMT labeling and mass spectrometry
MCF7 and MCF7 N1_KO cells expressing TurboID–V5–NONO from a doxycycline (DOX)-inducible promoter were cultivated in 15 cm culture dishes. TurboID–V5–NONO expression was turned on by cultivating the cells in the presence of 1 µg/ml DOX for 24 h before they were either left untreated or treated with SFN or grown in HBSS for 6 h. Biotin was added

for the last 30 min of the treatment period. Cells were washed five times with ice-cold PBS and nuclear extracts were prepared as described above. The extracts were incubated overnight at 4°C with streptavidin magnetic beads (Thermo Fisher Scientific, 88817) to isolate biotinylated proteins. The beads were thoroughly washed as described previously (Cho et al., 2020). Briefly, beads were washed twice with RIPA lysis buffer, once with 1 M KCl, once with 0.1 M Na$_2$CO$_3$, once with 2 M urea in 10 mM Tris-HCl pH 8.0 and twice with RIPA lysis buffer. Before being subjected to trypzination, the beads were washed twice in 2 M Urea in 50 mM Tris-HCl pH 7.5 and three times in 50 mM triethylammonium bicarbonate buffer (TEAB, Sigma-Aldrich, T7408). The immobilized proteins were pre-digested in 150 µl 50 mM TEAB, 1 mM CaCl$_2$ and 1 µg trypsin for 1 h and reduced in 5 mM DTT before being subjected to alkylation by 15 mM iodoacetamide. DTT was then added to a final concentration of 5 mM to quench iodoacetamide. Samples were then added 2 µg trypsin and digested overnight at 37°C in a Thermo Shaker at 800 rpm. Digested proteins were eluted from the beads, and the samples were acidified with trifluoroacetic acid (TFA) and C18 purified using Omix C18 pipette tips (Agilent, A57003100). Samples were then evaporated dry and dissolved in 20 µl 0.1% formic acid (FA). Peptide concentration was measured using a Nanodrop ONE spectrophotometer (Thermo Fisher Scientific). Samples were evaporated to remove acid and dissolved in 18 µl 200 mM TEAB for TMT labeling. Peptides were labeled with TMT using the TMT10plex™ Isobaric Label Reagent Set (Thermo Fisher Scientific, 90406) according to manufacturer's protocol. A bridge sample containing a pool of 2.5 µl of each sample was labeled with TMT 126. One 0.8 mg TMT vial was used to label four samples. After TMT labeling, TMT pool-samples were made by pooling 0.625 µg peptides pr sample from seven differently labeled samples – one sample from each cell subtype and treatment condition, and 0.625 µg of the bridge sample. Pooled samples were evaporated to remove acetonitrile (ACN) from the labeling process, dissolved in 20 µl 0.1% FA and measured by nanodrop. Peptides (0.5 µg) were fractionated using a 2–80% ACN gradient in 0.1% FA over 140 min at a flow rate of 300 nl/min. The separated peptides were analyzed using a Thermo Scientific Orbitrap Exploris 480 mass spectrometer. MS1 data were collected at 60k resolution over a $m/z$ range at 375–1200 using a maximum injection time of 50 ms. MS2 was collected at 45 K resolution at maximum injection time 105 ms. Raw data were analyzed using MaxQuant (version 2.4.9.0) with the integrated Andromeda search engine. An isobaric match between runs search was performed. MS/MS data was searched against the current Uniprot Human database. A false discover ratio (FDR) of 0.01 was needed to give a protein identification. Perseus (version 2.0.11) was used for statistical analysis. To control for differential protein loading within a TMT 8-plex, the summed protein quantities were adjusted to be equal within an 8-plex. Data were then log2 transformed. Proteins with more than two missing values were removed from the analysis. Missing values were imputed by the random forest method using the missForest package in R (https://cran.r-project.org/web/packages/missForest/index.html). The bridge channel protein quantity was subtracted from each sample quantity to create a ratio to the bridge. Bridge sample, now 0, was removed and within each ten-plex. The median protein expression was centered at 0. A network diagram was generated using a BioVenn online tool (https://www.biovenn.nl). Gene enrichment analysis was prepared using the Metascape online tool (https://metascape.org/) (Zhou et al., 2019). The mass spectrometry proteomics data have been deposited to the ProteomeXchange Consortium via the PRIDE (Perez-Riverol et al., 2025) partner repository with the dataset identifier PXD063223.

## Statistical analysis

Quantitative data were analyzed with Graphpad software (Prism 10). Statistical significance was determined by one-way or two-way ANOVA followed by Dunnett's or Tukey's multiple comparison tests, or by unpaired two-tailed Student's $t$-tests. Data are considered significant when the $P \le 0.05$. The error bars in all experiment indicate s.d. All experiments were repeated at least three times.

## Acknowledgements
The authors thank Terje Johansen, UiT the Arctic University of Norway, for generous gifts of reagents and plasmids, and Mireia Nager for constructing the pCW57.1_TurboID_V5 plasmid. The Mass spectrometry-based proteomic analyses were performed by UiT Proteomics and Metabolomics Core Facility (PRiME). This facility is a member of the National Network of Advanced Proteomics Infrastructure (NAPI), which is funded by the Research Council of Norway INFRASTRUKTUR-program ( project number: 295910). The authors wish to acknowledge resources and support from the Advanced Microscopy Core Facility at UiT The Arctic University of Norway. We thank Eva Sjøttem and Ingvild Mikkola for critically reading the manuscript and for providing valuable comments and suggestions.

## Competing interests
The authors declare no competing or financial interests.

## Author contributions
Conceptualization: M.P.; Formal analysis: I.O., A.K., S.D.S., K.B.L., J.-A.B., E.K.; Funding acquisition: M.P.; Investigation: M.P., I.O., B.K.S., S.R.N., I.E.B., A.H., T.A.G., S.M.L.; Project administration: M.P.; Supervision: M.P.; Validation: I.O., B.K.S., S.R.N., I.E.B., A.H., T.A.G., S.M.L.; Visualization: M.P., I.O.; Writing – original draft: M.P., I.O.; Writing – review & editing: M.P.

## Funding
This work was supported by Northern Norway Regional Health Authority (Helse Nord RHF; HNF1546-20). Open Access funding was provided by UiT The Arctic University of Norway. Deposited in PMC for immediate release.

## Data and resource availability
The LC-MS/MS data have been deposited to ProteomeXchange Consortium via PRIDE with identifier PXD063223. All other relevant data and details of resources can be found within the article and its supplementary information. Further inquiries can be directed to the corresponding author.

## Peer review history
The peer review history is available online at https://journals.biologists.com/jcs/lookup/doi/10.1242/jcs.264115.reviewer-comments.pdf

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
