## [Peer Review File · Journal of Cell Science]

Stress-specific NONO interactomes reveal a key role of Hsp70 chaperone activity in regulation of paraspeckle formation

Isaac Odonkor, Birendra Kumar Shrestha, Stephanie Rose Nielsen, Athanasios Kournoutis, Ida Emilie Bjørlo, Saikat Das Sajib, Annica Hedberg, Toril Anne Grønset, Kenneth Bowitz Larsen, Jack-Ansgar Bruun, Erik Knutsen, Seyed Mohammad Lellahi and Maria Perander
DOI: 10.1242/jcs.264115

Editor: Megan King

Review timeline

Original submission:	30 April 2025
Editorial decision:	2 June 2025
First revision received:	23 September 2025
Editorial decision:	13 November 2025
Second revision received:	18 November 2025
Accepted:	20 November 2025

Original submission

First decision letter

MS ID#: jcs.264115

MS TITLE: NEAT1-dependent stress-specific NONO interactomes reveal a key role of Hsp70 chaperone activity in regulation of paraspeckle formation

AUTHORS: Maria Perander; Isaac Odonkor; Birendra Kumar Shrestha; Stephanie Rose Nielsen; Athanasios Kournoutis; Ida Emilie Bjørlo; Saikat Das Sajib; Annica Hedberg; Toril Anne Grønset; Kenneth Bowitz Larsen; Jack-Ansgar Brun; Erik Knutsen; Seyed Mohammad Lellahi

ARTICLE TYPE: Research Article

Dear Dr Perander,

We have now reached a decision on the above manuscript.

To see the reviewers' reports and a copy of this decision letter, please go to:

As you will see, the reviewers raise a number of substantial criticisms that prevent me from accepting the paper at this stage. They suggest, however, that a revised version might prove acceptable, if you can address their concerns. If you think that you can deal satisfactorily with the criticisms on revision, I would be pleased to see a revised manuscript. We would then return it to the reviewers. Please pay special attention to the first major point from Reviewer 1, namely the need to interrogate cells lacking NEAT1.

Reviewer 1

In the first part of the manuscript, the authors utilize a TurboID-fused NONO-expressing cell line to identify NONO-interacting proteins under stress conditions induced by SFN treatment and starvation. Through comparison between wild-type and NEAT1 knockout cells, they identify Hsp70

and Nop56 as NEAT1-dependent interactors of NONO. In the latter part of the manuscript, the authors observe that NONO proteins accumulate in the nucleoli under stress, and propose that their subsequent translocation back to the nucleoplasm during recovery—dependent on Hsp70 activity—is essential for paraspeckle formation. While the attempt to characterize stress-specific paraspeckle components based on NEAT1-dependent NONO interactions is potentially intriguing, it is unclear how Hsp70 contributes to the stress-induced changes in NONO localization. I am not convinced that the central claim of this paper is correct. Furthermore, many of the experiments rely on exogenous tagged proteins, which weakens the physiological relevance of the conclusions. Therefore, the authors need to perform additional experiments to substantiate their claims.

Major comments:

Figure 3D

To validate the TurboID-MS results, the interaction between NONO and Hsp70 should be analyzed in NEAT1 knockout cells. Without this validation, the subheading "NONO associates with Hsp70 in a NEAT1-dependent manner in response to stress" (Line 182) is not appropriate. In addition, co-immunoprecipitation experiments using endogenous NONO and Hsp70 proteins should be performed to strengthen the conclusion.

Figure 6

To support the claim that "they (paraspeckles) primarily assemble during recovery after Hsp70 has relocated from the nucleolus to the nucleoplasm" (Lines 262-263), VER should be added during the recovery phase, rather than before heat stress.

Furthermore, NEAT1_2 expression should be analyzed under VER treatment in Figure 6D. Since Figure 4D shows that NEAT1_2 expression is inhibited by VER, it is possible that this inhibition persists during the recovery phase and contributes to the observed phenotypes. If this is the case, the data could be interpreted in a number of ways.

Figures 6A, 6B and 7C

To conclude that mCherry-Hsp70 and mCherry-NONO localize to the nucleoli, co-immunostaining with established nucleolar markers is required.

Moreover, NONO signals are also observed diffusely in the nucleoplasm under SFN and VER treatment. Can the authors rule out the possibility that these nucleoplasmic NONO proteins contribute to paraspeckle formation?

Figure S3B

The presented immunoblots do not clearly demonstrate an increase in Hsp70 protein levels upon SFN treatment (Line 197). To support this claim, more convincing blot images along with quantitative analysis should be provided.

Minor comments:

Line 135

It should be stated at the first mention that starvation refers to HBSS treatment.

Figure 2

The blue circles in B and C, presumably indicating TurboID-NONO, are not explained. It would be preferable to normalize the data to the amount of immunoprecipitated NONO to ensure accurate interpretation.

Figure 3A, 3B

Proteins that are stress-dependently enriched only in NEAT1 KO cells may provide important clues for understanding paraspeckle function, and it would be better to discuss their potential significance.

For improved readability, the protein list should be organized in alphabetical order.

Figure 6A

The "small speckles in the nucleus" formed by SFPQ after heat shock (Line 259) are referred to as "paraspeckles" in Lines 262 and 264. However, it has not been confirmed whether these small speckles are indeed paraspeckles. Further validation is needed.

Figure S1

It would be beneficial to include data showing the upregulation of NEAT1 expression and the increase in paraspeckle formation upon SFN treatment.

Figure 2D

It is unclear whether the Venn diagram corresponds to MCF7 WT, MCF7 NEAT1 KO, or a combination of both. This should be clearly stated in the figure legend.

Line 261

"Thermal stress" should be revised to "heat stress" for consistency throughout the manuscript.

Statistical analysis

The Dunnett test is appropriate for multiple comparisons against a common control; however, in many of the figures, the assumptions required for its valid application do not appear to be met. Additionally, the term "Non-parametric Student's t-test" is contradictory. Also, "Student" should be capitalized.

Reviewer 2

SUMMARY OF THE ADVANCE MADE IN THIS PAPER AND ITS POTENTIAL SIGNIFICANCE TO THE FIELD

The authors have characterized and compared two stress-enriched interactomes of the essential paraspeckle protein NONO in both wild type and paraspeckle-deficient NEAT1 knockout cells. They identify Hsp70 among the stress-enriched, NEAT1-dependent partners of NONO. They show that proteotoxic stress-induced paraspeckle formation and NEAT1 expression are dependent on Hsp70 chaperone activity. They conclude from their data that both NONO and Hsp70 transiently translocate to the nucleolus during heat shock and that paraspeckle formation during recovery follows Hsp70-dependent relocation of NONO from the nucleolus to the nucleoplasm. This suggests an important role of Hsp70 in paraspeckle assembly and a possible link between the nuclear protein quality control system and paraspeckles.

This results about a role of hsp70 are novel and interesting, although a bit descriptive. Anyway they are convincing and the paper is well written.

SUGGESTIONS TO AUTHORS

Major comments :

1/ Upon Hsp70 inhibition, the NONO relocation into nucleoli seems to occur only in U2OS after SFN treatment, but occurs in HeLa after heat shock treatment. Apparently there are differences between the cell types. The NONO-interactome has been analyzed only in MCF7, whereas NONO localization in Figure 7 is analyzed in U2OS and HeLa. The question is why heat shock experiments were not made with MCF7 ? The interaction between NONO and Hsp70 should be confirmed in U2OS and HeLa.

2/It seems contradictory that Hsp70 is not localized in the paraspeckle upon stress in HeLa, whereas it should interact with NONO.

This contradiction may result from the use of different cell lines where the paraspeckle formation process may differ, due to the involvement of other partners. It should be discussed.

3/ If Hsp70 is a family (13 members, as mentioned line 77), which one of these proteins has been identified and which one is used in the fusion protein experiments ? It is not clearly mentioned. In the article the term « hsp70 » is used as for a single protein. Please clarify.

Minor comments :

Lines 134 to 137 : « in cells that were either treated with sulforaphane (SFN) (Lellahi et al., 2018) or subjected to starvation, two conditions that induce NEAT1 expression and paraspeckle formation in MCF7 cells (Fig. S1A,B,C). »

Fig S1 only presents data with starved cells.

First revision

Author response to reviewers' comments

Response to editor and reviewers

Dear Editor and Reviewers

Thank you for the thorough review of our manuscript, and for your overall positive and very motivating feedback. We highly appreciate your comments and suggestions. Please read our response and description of changes we have made to the manuscript below. A new version of the manuscript where all changes are highlighted with red letters is provided. Fig. 3A-C, Fig. 6D, and Fig. S1 are revised according to your suggestions. We have corrected “*NEAT1_2* maxima” to “*NEAT1_2* intensity” in FIG. 4B and Fig. S1. We have included three additional supplementary figures, so that the total number is now eighth (Fig. S1-S8). The original title of the manuscript extended the allowed maximal number of characters and has therefore been changed from “*NEAT1*-dependent stress-specific NONO interactomes reveals a key role of Hsp70 chaperone activity in regulation of paraspeckle formation” to “**Stress-specific NONO interactomes reveals a key role of Hsp70 chaperone activity in regulation of paraspeckle formation**”. An additional study by Kawakami et al. that was published after the original submission of this manuscript, has been cited in the Discussion section (Kawakami et al., 2025).

Comments from the Reviewers:

Reviewer 1: In the first part of the manuscript, the authors utilize a TurboID-fused NONO-expressing cell line to identify NONO-interacting proteins under stress conditions induced by SFN treatment and starvation. Through comparison between wild-type and NEAT1 knockout cells, they identify Hsp70 and Nop56 as NEAT1-dependent interactors of NONO. In the latter part of the manuscript, the authors observe that NONO proteins accumulate in the nucleoli under stress, and propose that their subsequent translocation back to the nucleoplasm during recovery—dependent on Hsp70 activity—is essential for paraspeckle formation. While the attempt to characterize stress-specific paraspeckle components based on NEAT1-dependent NONO interactions is potentially intriguing, it is unclear how Hsp70 contributes to the stress-induced changes in NONO localization. I am not convinced that the central claim of this paper is correct. Furthermore, many of the experiments rely on exogenous tagged proteins, which weakens the physiological relevance of the conclusions. Therefore, the authors need to perform additional experiments to substantiate their claims.

Our response:

Thank you for your comment and for addressing a very relevant and central subject from our manuscript. Nucleolar relocation of NONO upon stress is well-described also by others (Fox et al., 2005, Fox et al., 2002, Shav-Tal et al., 2005, Andersen et al., 2002, Trifault et al., 2022, Trifault et al., 2024, Yasuhara et al., 2022), and NONO has previously been shown to be retained in the nucleoli upon Hsp70 inhibition (Frottin et al., 2019). We don't think or show that Hsp70 is involved in stress-induced nucleolar translocation of NONO, but that its activity is required for relocation back to the nucleoplasm during recovery. We show this for endogenous NONO in U2OS cells (Fig. 7A,B) and for ectopic mCherry-NONO in HeLa Flp-In cells (Fig. 7C). We suggest that nucleolar retainment of NONO upon Hsp70 inhibition impairs paraspeckle formation during recovery from heat shock. However, we do agree that we can't exclude the possibility that reduced paraspeckle formation upon Hsp70 inhibition might be partially or perhaps mainly due to other mechanisms. VER-155008 for sure reduces *NEAT1* levels, and the effect seems to be particularly pronounced on the long *NEAT1_2* isoform. However, if NONO is retained in the nucleolus, the stability of *NEAT1_2* might be affected as less NONO is available in the nucleoplasm for binding to *NEAT1_2*. We propose this but agree that there is a possibility that a separate pool of NONO that remains in the nucleoplasm is engaged in paraspeckle formation. Most likely NONO actively shuttles between the nucleolus and the nucleoplasm. We can mention that VER-155008 does not affect the *NEAT1* promoter as measured by reporter gene assays (both luciferase activity and *luciferase* expression by RT-qPCR). We decided not to include these data in the current manuscript. Generally, we do agree that some of our statements should be moderated, and we have done so throughout the

manuscript.

Major comments:

Figure 3D

To validate the TurboID-MS results, the interaction between NONO and Hsp70 should be analyzed in NEAT1 knockout cells. Without this validation, the subheading "NONO associates with Hsp70 in a NEAT1-dependent manner in response to stress" (Line 182) is not appropriate. In addition, co-immunoprecipitation experiments using endogenous NONO and Hsp70 proteins should be performed to strengthen the conclusion.

Our response:

Thank you for this highly relevant comment. We have repeatedly tried to perform co-immunoprecipitation experiments of endogenous NONO and Hsp70 in wild type and *NEAT1* knockout MCF7 cells. However, we have not succeeded. We have tried to pull down both Hsp70 and NONO using altogether 4 different antibodies and looked for co-immunoprecipitation of the other protein. We suspect that this is due to antibody-related technical issues. By inspecting the literature, we do see that similar studies mostly employ epitope-tagged proteins. This is exemplified in (Yu et al., 2021). The subheading "NONO associates with Hsp70 in a *NEAT1*-dependent manner in response to stress" is primarily based on results from the proximity proteomics experiments, and "*NEAT1*-dependent" refers to differences between wild type and *NEAT1* knockout cells in terms of enrichment of Hsp70 in NONO complexes. Since our data are based on biotin ligase proximity labeling, we don't necessarily think that *NEAT1* is involved in a direct interaction between NONO and Hsp70. In contrast, we do think NONO and Hsp70 come into proximity to each other in the nucleolus and for unknown reasons, *NEAT1* (and possibly paraspeckles) seem to promote this. Nevertheless, we agree that we can moderate our statement and "NONO associates with Hsp70 in a *NEAT1*-dependent manner in response to stress" has now been changed to "**NONO associates with Hsp70 in response to stress**"

Figure 6

To support the claim that "they (paraspeckles) primarily assemble during recovery after Hsp70 has relocated from the nucleolus to the nucleoplasm" (Lines 262-263), VER should be added during the recovery phase, rather than before heat stress.

Furthermore, NEAT1_2 expression should be analyzed under VER treatment in Figure 6D. Since Figure 4D shows that NEAT1_2 expression is inhibited by VER, it is possible that this inhibition persists during the recovery phase and contributes to the observed phenotypes. If this is the case, the data could be interpreted in a number of ways.

Our response:

Thank you for this highly relevant comment. This is for sure a very interesting point that addresses a central part of our paper. We think relocation of NONO from the nucleolus to the nucleoplasm is the critical incidence and prerequisite for formation of paraspeckles during recovery, and Hsp70 activity is required for this. Although we feel confident that our interpretation is well-founded on our experimental results, we totally agree that there can be alternative explanations. We have indeed shown that VER-155008 inhibits basal and SFN-induced expression of *NEAT1* in HeLa cells (Fig. 4) and have now included *NEAT1* expression data in FIG. 6D showing that adding VER-155008 before HS also seriously reduced HS-induced *NEAT1*. In Fig 4D, we show that VER-155008 seemed to have a stronger effect on *NEAT1_2* than total *NEAT1* expression, and as mentioned above, we have unpublished data showing that the inhibitor does not affect transcription from the *NEAT1* promoter. This could suggest that inhibition of Hsp70 compromises the stability of the *NEAT1_2* isoform, possibly through retaining NONO in the nucleolus.

We have tried to add VER-155008 at different time points during recovery from HS. This became very technically challenging for us. For unknown reasons, we experienced software issues with the CellDiscoverer 7 automated widefield microscope used in time lapse studies when trying to add VER-155008 during the experiment. We then conducted confocal imaging experiments on fixed cells, but lost resolution and signal-to-noise when trying to analyze GFP-SFPQ-containing paraspeckles. Finally, we ended up conducting *NEAT1_2*-specific RT-qPCR experiments. We noticed

that adding VER-155008 during the first two hours of recovery led to similar nucleolar retention of Hsp70 at 6 h recovery as when the compound was added before HS. We postulated based on the results shown in Fig. 4D that 2-hour incubation in the presence of VER-155008 was enough to give maximum reduction in *NEAT1_2* levels. We therefore added VER-155008 3 h and 4 h after HS, and measured *NEAT1_2* expression at 6 h after HS. The results are shown below. We do see that HS-induced *NEAT1_2* expression is less sensitive to VER-155008 when added during the recovery, but it is hard to make firm conclusions from these data, and we have decided not to include it in the revised manuscript. However, as mentioned above, we have moderated some of our statements throughout the manuscript.

Figures 6A, 6B and 7C

To conclude that mCherry-Hsp70 and mCherry-NONO localize to the nucleoli, co-immunostaining with established nucleolar markers is required.

Moreover, NONO signals are also observed diffusely in the nucleoplasm under SFN and VER treatment. Can the authors rule out the possibility that these nucleoplasmic NONO proteins contribute to paraspeckle formation?

Our response:

Thank you for this highly relevant comment. We have now performed immunostaining with an antibody towards nucleolin confirming that mCherry-Hsp70 and mCherry-NONO translocate to the nucleoli in response to HS and are retained in the structures in VER-155008-treated cells (Fig. S5 and Fig. S8). We can't rule out that NONO proteins that remain nucleoplasmic during SFN and VER-155008 treatment contribute to paraspeckle formation. NONO for sure shuttles between the nucleolus and the nucleoplasm. Because of this, we have moderated some of our statements.

Figure S3B

The presented immunoblots do not clearly demonstrate an increase in Hsp70 protein levels upon SFN treatment (Line 197). To support this claim, more convincing blot images along with quantitative analysis should be provided.

Our response:

Thank you so much for noticing this. We never intended to claim that Hsp70 protein levels increased in response to SFN in any of the cell lines, but made a mistake during writing of the manuscript that we missed during internal proofreading. The data shown in Fig S3 are now discussed properly. We apologize for this.

Line 135

It should be stated at the first mention that starvation refers to HBSS treatment.

Our response:

Thank you. We totally agree and have specified this in the text.

Figure 2

The blue circles in B and C, presumably indicating TurboID-NONO, are not explained. It would be preferable to normalize the data to the amount of immunoprecipitated NONO to ensure accurate interpretation.

Our response:

Thank you. The blue circles are indeed indicating the TurboID protein (assumingly as a TurboID-NONO fusion). We have now explained this in the figure legend.

Regarding the normalization to the amount of TurboID-NONO: This has been a long discussion within the team, and we have read appropriate literature and discussed with experienced personnel at the proteomic core facility. SFN reduces TurboID-NONO levels in wild type MCF7 cells, and we must admit that this has caused some frustrations. However, to normalize all the purified biotinylated proteins in the extracts to TurboID-NONO levels will not be correct as variations in TurboID-NONO do not reflect the total amount of proteins in the samples, and TurboID-NONO is not immunoprecipitated as such. To overcome the problem with reduced TurboID-NONO levels in treated versus untreated cells, we have only focused proteins that become enriched during stress. The fact that these candidates become enriched even though the amount of the bait is reduced, strongly suggests that they are indeed bona fide stress-enriched NONO partners. The low number of detected proteins that become enriched upon SFN in wild type cells is most likely due to the reduced expression of TurboID-NONO, and represents a limitation of this study. This issue has been carefully addressed in the manuscript.

Figure 3A, 3B

Proteins that are stress-dependently enriched only in NEAT1 KO cells may provide important clues for understanding paraspeckle function, and it would be better to discuss their potential significance.

For improved readability, the protein list should be organized in alphabetical order.

Our response:

Thank you and we totally agree. However, our decision to only focus on Hsp70 in this manuscript is related to the issue described above. For unknown (and potentially interesting) reasons, SFN treatment reduces TurboID-NONO levels in wild type cells, but not, or to a much lesser extent, in *NEAT1* knockout cells. Because of this, the differences in the number of enriched proteins between the two cell lines might not reflect the true situation. We therefore decided to focus only on proteins that became enriched in wild type but not in *NEAT1* KO cells, and not vice versa. We have organized the proteins in alphabetical order and agree that this improved the readability.

Figure 6A

The "small speckles in the nucleus" formed by SFPQ after heat shock (Line 259) are referred to as "paraspeckles" in Lines 262 and 264. However, it has not been confirmed whether these small speckles are indeed paraspeckles. Further validation is needed.

Our response:

Thank you for this highly relevant comment. We agree that this was indeed written unclearly, and we apologize for that. We don't believe that the initially formed small speckles are paraspeckles, but that bona fide paraspeckles form later during recovery from heat shock. We have clarified this in the text and included GFP- and *NEAT1_2*-specific co-immuno-FISH experiments showing that GFP-SFPQ colocalizes with *NEAT1_2* in the larger bodies that form approximately after 180 min of recovery (Fig. S6).

Figure S1

It would be beneficial to include data showing the upregulation of NEAT1 expression and the increase in paraspeckle formation upon SFN treatment.

Our response:

Thank you for your suggestion. We initially decided not to include it as we have shown this in a previous publication. However, we totally agree that by including it, we improve the flow of the current manuscript. The data are presented in Figure S1.

Figure 2D

It is unclear whether the Venn diagram corresponds to MCF7 WT, MCF7 NEAT1 KO, or a combination of both. This should be clearly stated in the figure legend.

Our response:

Thank you for your suggestion. We have now clarified this in the figure legend.

Line 261

"Thermal stress" should be revised to "heat stress" for consistency throughout the manuscript.

Our response:

Thank you. We have revised this according to your suggestion.

Statistical analysis

The Dunnett test is appropriate for multiple comparisons against a common control; however, in many of the figures, the assumptions required for its valid application do not appear to be met.

Additionally, the term "Non-parametric Student's t-test" is contradictory. Also, "Student" should be capitalized.

Our response:

Thank you and we totally agree. We have now carefully gone through all the analyses and have realized that the Tukey's multiple comparison test was indeed used for statistical analyses of imaging data in Fig. 4B, Fig. 7D, and Fig. S4B, and RT-qPCR data in FIG. 4C, 6D, and S4C. This has now been corrected in the text and figure legends. The "non-parametric Student's t-test" is a typo and is corrected to unpaired Student's t-test. We have capitalized "Student's" and corrected a misspelling of Dunnett throughout the manuscript. We thank you again for noticing this.

Reviewer 2: SUMMARY OF THE ADVANCE MADE IN THIS PAPER AND ITS POTENTIAL SIGNIFICANCE TO THE FIELD

The authors have characterized and compared two stress-enriched interactomes of the essential paraspeckle protein NONO in both wild type and paraspeckle-deficient NEAT1 knockout cells. They identify Hsp70 among the stress-enriched, NEAT1-dependent partners of NONO. They show that proteotoxic stress-induced paraspeckle formation and NEAT1 expression are dependent on Hsp70 chaperone activity. They conclude from their data that both NONO and Hsp70 transiently translocate to the nucleolus during heat shock and that paraspeckle formation during recovery follows Hsp70-dependent relocation of NONO from the nucleolus to the nucleoplasm. This suggests an important role of Hsp70 in paraspeckle assembly and a possible link between the nuclear protein quality control system and paraspeckles. This results about a role of hsp70 are novel and interesting, although a bit descriptive. Anyway they are convincing and the paper is well written.

Our response:

Thank you for your overall positive and motivating feedback.

SUGGESTIONS TO AUTHORS

Major comments :

1/ Upon Hsp70 inhibition, the NONO relocation into nucleoli seems to occur only in U2OS after SFN treatment, but occurs in HeLa after heat shock treatment. Apparently there are differences between the cell types. The NONO-interactome has been analyzed only in MCF7, whereas NONO localization in Figure 7 is analyzed in U2OS and HeLa. The question is why heat shock experiments were not made with MCF7 ?

Our response:

Thank you for your very relevant comment. There are for sure cell line-dependent differences in terms of detecting endogenous NONO in the nucleolus upon VER-155008 and SFN by immunofluorescent staining (IF). The reason for this is unclear. The U2OS cells express much lower basal levels of *NEAT1* compared to the other cell lines. Could that contribute to a higher degree of nucleolar retention of NONO upon SFN and VER? Does NONO to a larger extent shuttle between the nucleolus and the nucleoplasm in U2OS compared to HeLa and MCF7 cells? Are U2OS cells more prone to nuclear protein aggregation than the others? Anyhow, nucleolar relocalization of NONO in response to cellular stress is well documented by others), which supports our results (Fox et al., 2005, Fox et al., 2002, Trifault et al., 2022, Trifault et al., 2024, Shav-Tal et al., 2005, Yasuhara et al., 2022, Andersen et al., 2002, Frottin et al., 2019).

The reason for changing cellular system from MCF7 to HeLa Flp-In for the imaging experiments, was to establish cell lines that allowed simultaneous real-time imaging analyses of paraspeckle formation and Hsp70 or NONO localization. It has previously been demonstrated that it is technically challenging to detect Hsp70 in condensates by conventional IF, leaving imaging analyses of a fluorescently tagged protein as the only option for such studies (Yu et al., 2021). The HeLa Flp-In system is well-established in our lab and offers a possibility to moderately overexpress a gene of interest in a doxocycline-inducible manner. We successfully established a stable HeLa Flp-In cell line that expresses endogenous SFPQ fused to GFP, which allowed real-time monitoring of SFPQ-containing paraspeckles. We inserted either mCherry-Hsp70 or mCherry-NONO into FRT site, making it possible to simultaneously monitor paraspeckle-formation and mCherry-Hsp70 or mCherry-NONO localization. Unfortunately, we were not able to develop a similar system in MCF7 cells. Although, this might impact the readability and flow of the study to some extent, we believe that working with different cell line models also represents a strength to our studies.

The interaction between NONO and Hsp70 should be confirmed in U2OS and HeLa.

Our response:

Thank you for your suggestion. In the original manuscript, we showed that both ectopic Myc-tagged NONO and endogenous NONO co-immunoprecipitated with GFP-Hsp70 in HeLa Flp-In cells (FIG. 3D). We have stated this more clearly in the revised manuscript. Moreover, as described above, we have used a lot of effort during the revision period trying to coimmunoprecipitate endogenous proteins in MCF7 cells. However, we have unfortunately not succeeded.

2/It seems contradictory that Hsp70 is not localized in the paraspeckle upon stress in HeLa, whereas it should interact with NONO.

This contradiction may result from the use of different cell lines where the paraspeckle formation process may differ, due to the involvement of other partners. It should be discussed.

Our response:

Thank you for your very relevant comment. Based on the proteomics data, we originally hypothesized that the enrichment of Hsp70 in the stress-induced NONO interactomes, particularly in MCF7 wild type cells, was due to relocation of Hsp70 to paraspeckles. However, we have not been able to detect neither endogenous (data not shown) nor fluorescently tagged ectopic Hsp70 in paraspeckles. We therefore believe that NONO and Hsp70 do not colocalize in the paraspeckles, but that both proteins relocate to the nucleoli upon stress. The nucleolus plays a critical role in nuclear proteostasis and upon proteotoxic stress, aggregation-prone nuclear proteins relocate to the nucleolus where Hsp70 keeps them in a state that allows efficient refolding during recovery (Frottin et al., 2019). We must admit that during the course of the experiments included in this manuscript, the storyline changed from being a study of paraspeckle functions to a study of the impact of Hsp70 on paraspeckle formation.

3/ If Hsp70 is a family (13 members, as mentioned line 77), which one of these proteins has been identified and which one is used in the fusion protein experiments? It is not clearly mentioned. In the article the term « hsp70 » is used as for a single protein. Please clarify.

Our response:

Thank you for your comment, which we can see is very relevant. All our studies are focused on the canonical stress-induced *HSPA1A*- and *HSPA1B*-encoded Hsp70 proteins that are nearly identical. They are not differentiated in the proteomic data. We have therefore changed *HSPA1B* to *HSPA1A;HSPA1B* in the lists of proteins presented in figures 3A-3C. The GFP-Hsp70 protein is encoded by *HSPA1B* fused to the GFP gene. We will clarify that. Regarding the RT-qPCR experiment described in FigS3A, we are specifically amplifying *HSPA1A*, and not *HSPA1B*, as originally indicated in the figure. We apologize for this mistake, and we have now changed it.

Minor comments :

Lines 134 to 137 : « in cells that were either treated with sulforaphane (SFN) (Lellahi et al., 2018) or subjected to starvation, two conditions that induce *NEAT1* expression and paraspeckle formation in MCF7 cells (Fig. S1A,B,C). »

Fig S1 only presents data with starved cells.

Our response:

Thank you. We have now also included data for SFN-treated cells in Fig. S1.

REFERENCES

- ANDERSEN, J. S., LYON, C. E., FOX, A. H., LEUNG, A. K., LAM, Y. W., STEEN, H., MANN, M. & LAMOND, A. I. 2002. Directed proteomic analysis of the human nucleolus. *Curr Biol*, 12, 1-11.
- FOX, A. H., BOND, C. S. & LAMOND, A. I. 2005. P54nrb forms a heterodimer with PSP1 that localizes to paraspeckles in an RNA-dependent manner. *Mol Biol Cell*, 16, 5304-15.
- FOX, A. H., LAM, Y. W., LEUNG, A. K., LYON, C. E., ANDERSEN, J., MANN, M. & LAMOND, A. I. 2002. Paraspeckles: a novel nuclear domain. *Curr Biol*, 12, 13-25.
- FROTTIN, F., SCHUEDER, F., TIWARY, S., GUPTA, R., KORNER, R., SCHLICHTHAERLE, T., COX, J., JUNGSMANN, R., HARTL, F. U. & HIPPEL, M. S. 2019. The nucleolus functions as a phase-separated protein quality control compartment. *Science*, 365, 342-347.
- KAWAKAMI, Y., IGUCHI, Y., LI, J., AMAKUSA, Y., YOSHIMURA, T., CHIKUCHI, R., YOKOI, S., IIDA, M., RIKU, Y., IWASAKI, Y., HIROSE, T., NAKAGAWA, S. & KATSUNO, M. 2025. Downregulation of *NEAT1* due to loss of TDP-43 function exacerbates motor neuron degeneration in amyotrophic lateral sclerosis. *Brain Commun*, 7, fcaf261.
- SHAV-TAL, Y., BLECHMAN, J., DARZACQ, X., MONTAGNA, C., DYE, B. T., PATTON, J. G., SINGER, R. H. & ZIPORI, D. 2005. Dynamic sorting of nuclear components into distinct nucleolar caps during transcriptional inhibition. *Mol Biol Cell*, 16, 2395-413.
- TRIFAULT, B., MAMONTOVA, V. & BURGER, K. 2022. In vivo Proximity Labeling of Nuclear and Nucleolar Proteins by a Stably Expressed, DNA Damage-Responsive NONO-APEX2 Fusion Protein. *Front Mol Biosci*, 9, 914873.
- TRIFAULT, B., MAMONTOVA, V., COSSA, G., GANSKIH, S., WEI, Y., HOFSTETTER, J., BHANDARE, P., BALUAPURI, A., NIETO, B., SOLVIE, D., ADE, C. P., GALLANT, P., WOLF, E., LARSEN, D. H., MUNSCHAUER, M. & BURGER, K. 2024. Nucleolar detention of NONO shields DNA double-strand breaks from aberrant transcripts. *Nucleic Acids Res*, 52, 3050-3068.
- YASUHARA, T., XING, Y. H., BAUER, N. C., LEE, L., DONG, R., YADAV, T., SOBERMAN, R. J., RIVERA, M. N. & ZOU, L. 2022. Condensates induced by transcription inhibition localize active chromatin to nucleoli. *Mol Cell*, 82, 2738-2753 e6.
- YU, H., LU, S., GASIOR, K., SINGH, D., VAZQUEZ-SANCHEZ, S., TAPIA, O., TOPRANI, D., BECCARI, M. S., YATES, J. R., 3RD, DA CRUZ, S., NEWBY, J. M., LAFARGA, M., GLADFELTER, A. S., VILLA, E. & CLEVELAND, D. W. 2021. HSP70 chaperones RNA-free TDP-43 into anisotropic intranuclear liquid spherical shells. *Science*, 371.

Second decision letter

MS ID#: jcs.264115R1

MS TITLE: Stress-specific NONO interactomes reveal a key role of Hsp70 chaperone activity in regulation of paraspeckle formation

AUTHORS: Maria Perander; Isaac Odonkor; Birendra Kumar Shrestha; Stephanie Rose Nielsen; Athanasios Kournoutis; Ida Emilie Bjørlo; Saikat Das Sajib; Annica Hedberg; Toril Anne Grønset; Kenneth Bowitz Larsen; Jack-Ansgar Brun; Erik Knutsen; Seyed Mohammad Lellahi
ARTICLE TYPE: Research Article

Dear Dr Perander,

Thank you for your patience. We apologize that the review of your revised manuscript has taken such a long period of time as we tried to engage both prior reviewers of the first submission. We have elected to move forward despite not receiving one of the reviews.

As you will see, the reviewer who assessed the revised manuscript continues to raise some critical points that will require edits to the text of your manuscript. If you could please address these comments by responding to the three caveats with changes to the text then I will be able to accept your paper without further delay.

Reviewer 1

I am not satisfied with the revised manuscript because the authors have not addressed the most important issues raised in my previous review.

Point 1.

The original title was "NEAT1-dependent stress-specific NONO interactomes reveal a key role of Hsp70 chaperone activity in the regulation of paraspeckle formation," and the original subheading was "NONO associates with Hsp70 in a NEAT1-dependent manner in response to stress." The phrase "NEAT1-dependent" was repeatedly emphasized throughout the initial manuscript, clearly indicating that NEAT1 dependency was intended as the central claim of the paper. Therefore, I recommended that the interaction between NONO and Hsp70 should be examined in NEAT1 knockout cells in Figure 3. However, the authors' response – "we don't necessarily think that NEAT1 is involved in a direct interaction between NONO and Hsp70" – suggests that the central claim has shifted. Moreover, despite having access to NEAT1 KO cells, the authors have still not performed the appropriate experiment.

Point 2.

Regarding Figure 3D, the following statement must be explicitly included in the manuscript: "We have tried to pull down both Hsp70 and NONO using four different antibodies and examined the co-immunoprecipitation of the other protein. We suspect that this failure is due to antibody-related technical issues."

Point 3.

My previous comment on Figure S3B was:

"The presented immunoblots do not clearly demonstrate an increase in Hsp70 protein levels upon SFN treatment (Line 197). To support this claim, more convincing blot images along with quantitative analysis should be provided."

The authors responded:

"We never intended to claim that Hsp70 protein levels increased in response to SFN in any of the cell lines, but made a mistake during the writing of the manuscript that we missed during internal proofreading. The data shown in Fig S3 are now discussed properly. We apologize for this." If so, the authors should clarify why HSPA1A mRNA levels increase while Hsp70 protein levels appear unchanged. Does the antibody used detect multiple HSP70 family members?

Second revision

Author response to reviewers' comments

Response to editor and reviewers, Nov 17, 2025

Dear Editor and Reviewers

Thank you once again for the thorough review of our manuscript. We have made some changes to the text according to the concerns raised by the reviewer. This is highlighted in green. Please see our comments regarding the final questions raised by the reviewer below.

Please note that for the RT-qPCR data presented in Figs. 4C, 4D, 6D, S1A, S1D, S3A, and S4C, we show the data spread in the final submitted versions of the figures. A couple of minor changes are made in the figure legends for Fig. 1 and Fig. 4 (highlighted in green), and Fig S3 and Fig.S4.

Point 1.

The original title was "NEAT1-dependent stress-specific NONO interactomes reveal a key role of Hsp70 chaperone activity in the regulation of paraspeckle formation," and the original subheading was "NONO associates with Hsp70 in a NEAT1-dependent manner in response to stress." The phrase "NEAT1-dependent" was repeatedly emphasized throughout the initial manuscript, clearly indicating that NEAT1 dependency was intended as the central claim of the paper. Therefore, I recommended that the interaction between NONO and Hsp70 should be examined in NEAT1 knockout cells in Figure 3. However, the authors' response – "we don't necessarily think that NEAT1 is involved in a direct interaction between NONO and Hsp70" – suggests that the central claim has shifted. Moreover, despite having access to NEAT1 KO cells, the authors have still not performed the appropriate experiment.

Our response:

Thank you for your comment. The central claim of our paper has always been that **Hsp70 activity is required for paraspeckle formation**, which is to our knowledge a novel and highly topical finding. Our data add another layer of knowledge to regulatory roles for protein chaperones in biomolecular condensate formation, which might be highly relevant in human diseases such as neurodegenerative disorders. In our previous response to the reviewers, we honestly wrote "We must admit that during the course of the experiments included in this manuscript, the storyline changed from being a study of paraspeckle functions to a study of the impact of Hsp70 on paraspeckle formation". This is the nature of basic experimental research.

We modified the subheading "NONO associates with Hsp70 in a *NEAT1*-dependent manner in response to stress" as response to your comments after some consideration. The original subheading directly referred to our proteomic data showing that Hsp70 became enriched in stress-induced NONO complexes in wild type cells but not in *NEAT1* knockout cells. We do think that *NEAT1* through an unknown mechanism facilitates complex formation between NONO and Hsp70 upon proteotoxic stress in vivo, possibly by facilitating nucleolar translocation of NONO or through modulating Hsp70 dynamics. This is currently the focus of ongoing research in the lab. We chose to edit the subheading as we do agree that "*NEAT1*-dependent" could be interpreted as *NEAT1* was required for direct protein:protein interaction between NONO and Hsp70, which we have not shown. As for the title of the manuscript, the original title was too long according to the criteria set by Journal of Cell Science, and for us the most important finding to highlight is that **Hsp70 activity is required for paraspeckle formation**.

When performing coimmunoprecipitation experiments for the revised manuscript trying to show interaction between endogenous NONO and Hsp70, we always included both wild type MCF7 and MCF7 *NEAT1* knockout cells. Unfortunately, these experiments did not turn out successfully. The coimmunoprecipitation experiments of overexpressed proteins described in Fig. 3D, were done in HeLa cells that are generally easier to transfect (at least in our hands) than MCF7 cells. We did not include any studies in HeLa *NEAT1* knockout cells in this manuscript.

All in all, we understand the questions raised by you as they are indeed relevant. We have tried to conduct the suggested experiments and made some modifications to the text. We agree that we should state more clearly that we have not shown that *NEAT1* is required for direct interaction between NONO and Hsp70. This has now been included in the Discussion section.

Point 2.

Regarding Figure 3D, the following statement must be explicitly included in the manuscript:

"We have tried to pull down both Hsp70 and NONO using four different antibodies and examined the co-immunoprecipitation of the other protein. We suspect that this failure is due to antibody-related technical issues."

Our response:

Thank you for your comment. We now have included this in the Discussion part of the manuscript.

Point 3.

My previous comment on Figure S3B was:

"The presented immunoblots do not clearly demonstrate an increase in Hsp70 protein levels upon SFN treatment (Line 197). To support this claim, more convincing blot images along with quantitative analysis should be provided."

The authors responded:

"We never intended to claim that Hsp70 protein levels increased in response to SFN in any of the cell lines, but made a mistake during the writing of the manuscript that we missed during internal proofreading. The data shown in Fig S3 are now discussed properly. We apologize for this."

If so, the authors should clarify why HSPA1A mRNA levels increase while Hsp70 protein levels appear unchanged. Does the antibody used detect multiple HSP70 family members?

Our response:

Thank you for your comment. Our antibody does recognize Hsc70 encoded by the *HSPA8* gene. However, we do see an induction of *HSPA8* by SFN at the mRNA level (data not shown). We think the most likely explanation for our results, is that a 6-hour incubation period in the presence of SFN is not long enough to see any major changes in the protein levels. We haven't followed this up as the purpose for showing this in the current study was to demonstrate that the Hsp70 protein levels are equal in wild type and *NEAT1* knockout cells. We agree that we should comment on this in the text, which has now been included.

Third decision letter

MS ID#: jcs.264115R2

MS Title: Stress-specific NONO interactomes reveal a key role of Hsp70 chaperone activity in regulation of paraspeckle formation

Authors: Maria Perander; Isaac Odonkor; Birendra Kumar Shrestha; Stephanie Rose Nielsen; Athanasios Kournoutis; Ida Emilie Bjørlo; Saikat Das Sajib; Annica Hedberg; Toril Anne Grønset; Kenneth Bowitz Larsen; Jack-Ansgar Brun; Erik Knutsen; Seyed Mohammad Lellahi
Article Type: Research Article

Dear Dr Perander,

I am happy to tell you that your manuscript has been accepted for publication in Journal of Cell Science, pending standard publication integrity checks.